# Let Features Decide Their Own Solvers: Hybrid Feature Caching for Diffusion Transformers

**Shikang Zheng**[1,2]  **Guantao Chen**[1]  **Qinming Zhou**[1,3]  **Yuqi Lin**[1]  **Lixuan He**[1,3]
**Chang Zou**[1]  **Peiliang Cai**[1]  **Jiacheng Liu**[1]  **Linfeng Zhang**[1†]

[1]Shanghai Jiao Tong University  [2]South China University of Technology  [3]Tsinghua University

## Abstract

Diffusion Transformers offer state-of-the-art fidelity in image and video synthesis, but their iterative sampling process remains a major bottleneck due to the high cost of transformer forward passes at each timestep. To mitigate this, feature caching has emerged as a training-free acceleration technique that reuses hidden representations. However, existing methods often apply a uniform caching strategy across all feature dimensions, ignoring their heterogeneous dynamic behaviors. Therefore, we adopt a new perspective by modeling hidden feature evolution as a mixture of ODEs across dimensions, and introduce **HyCa**, a Hybrid ODE solver inspired caching framework that applies dimension-wise caching strategies. HyCa achieves near-lossless acceleration across diverse tasks and models, including $5.55\times$ speedup on FLUX, $5.56\times$ speedup on HunyuanVideo, $6.24\times$ speedup on Qwen-Image and Qwen-Image-Edit without retraining. **Our Project Page.**

## 1 Introduction

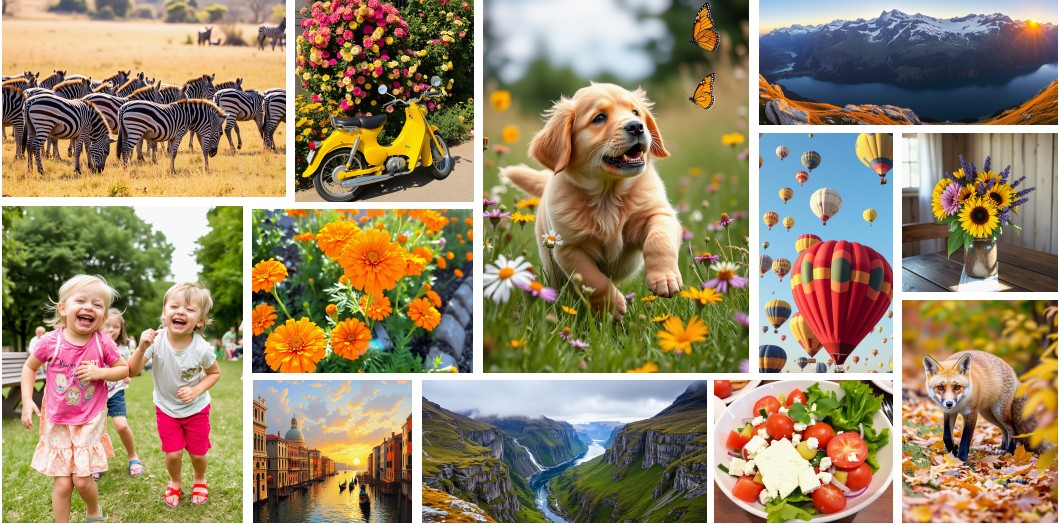

Figure 1: Images generated on Qwen-Image using HyCa combined with distillation at **12×** speedup.

Diffusion Transformers (DiTs) have recently achieved impressive success across image and video generation tasks, demonstrating strong modeling capacity and generation quality. However, the iterative nature of diffusion sampling presents a significant bottleneck, as each output demands multiple transformer passes. This high computational cost hinders deployment in scenarios with strict latency or resource constraints, driving the ongoing research on efficient inference methods.

To address this challenge, two primary acceleration directions have emerged: reducing the total number of sampling steps via algorithmic advancements (Lu et al., 2022a), and lowering the cost of each step through architectural optimization (Yuan et al., 2024; Zhao et al., 2024). Among these,

---

[†]Corresponding author.

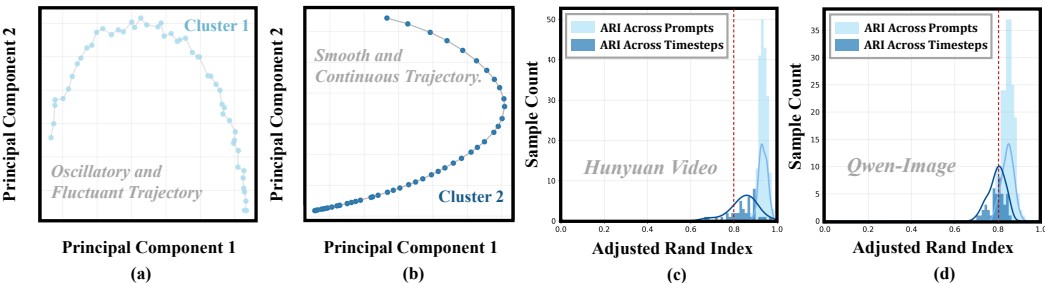

Figure 2: **Feature trajectory clusters and stability of assignments.** (a–b) Cluster 1 shows oscillatory trajectories while Cluster 2 shows smooth ones. (c–d) ARI distributions on Hunyuan Video and Qwen-Image exceed 0.8 in most cases, confirming stable and consistent cluster assignments across prompts and timesteps. An ARI above 0.8 indicates strong agreement and high clustering reliability.

training-free feature caching has emerged as a promising solution. It exploits the temporal coherence of hidden representations by reusing features, thereby reducing redundant computation. Early works such as DeepCache (Ma et al., 2024) demonstrated the feasibility of this idea in U-Net backbones, recent methods such as FORA (Selvaraju et al., 2024), ToCa (Zou et al., 2024a), TaylorSeer (Liu et al., 2025a) extended caching to transformer-based architectures and showed that feature caching can be effectively viewed as solving the temporal evolution of hidden features. Despite progress, current approaches are still limited in critical ways.

Existing methods implicitly assume that all hidden dimensions evolve under a single, unified system. However, this assumption is untenable in DiTs, where the feature space is high-dimensional and exhibits complicated behaviors. Such complexity is unlikely to be captured by a single process. To further investigate, we analyze how each feature dimension changes over timesteps and group them into clusters based on their dynamics. As shown in Fig. 2(a), some dimensions fluctuate sharply with oscillatory patterns, indicating stiffness or multimodal behavior, while others evolve smoothly and predictably, reflecting stable dynamics on Fig. 2(b). These observations suggest that the feature space of DiTs is better described as a complex system, where different groups of dimensions follow distinct temporal patterns, highlighting the need for tailored solvers rather than a one-size-fits-all approach.

Therefore, we introduce **HyCa**, a hybrid caching framework that models hidden feature evolution as a mixture of ODEs and applies suitable solvers for every dimension. Hyca begins with unsupervised clustering, grouping dimensions with similar dynamic behaviors, and modeling them into a shared ODE. Then, HyCa assigns the most suitable solver to each cluster. Normally, identifying the best solver would require running inference on a large set of images and comparing quantitative metrics. However, surprisingly, we found that cluster assignments are highly stable across resolutions, timesteps, and even prompts. As shown in Fig. 2(c)(d), this invariance allows us to evaluate solver performance on a **single prompt** at a **single timestep** to reliably identify the best solver, achieving results comparable to large-scale evaluation. Thus, with *"One-Time Choosing"* performed offline for each model, *"All-Time Solving"* becomes possible without any additional cost during inference.

HyCa provides robust and adaptive feature prediction across diverse tasks and architectures. Without retraining, it achieves near-lossless acceleration of **5.56×** on FLUX and Hunyuan Video, **6.24×** on Qwen-Image and Qwen-Image-Edit. Moreover, it is also fully compatible with distillation, reaching up to **24.4×** speedup on FLUX and **12.2×** on Qwen-Image while maintaining strong image quality. In summary, our main contributions are:

- **Heterogeneous Feature Dynamics.** We show that feature dimensions in DiTs do not follow a single unified system but exhibit heterogeneous dynamic behaviors that are better described as a mixture of ODEs. Through dynamics clustering analysis across multiple settings, we further reveal that these cluster's distributions are consistent and input-invariant.

- **HyCa Framework.** Inspired by hybrid ODE solvers in numerical analysis, we propose HyCa, a training-free framework that groups feature dimensions by their dynamics and automatically assigns the most suitable solver to each group, with minimal overhead.

- **Outstanding Performance.** We evaluate HyCa across diverse architectures and tasks, including Drawbench on FLUX and Qwen-Image, Vbench on HunyuanVideo, GEdit-Bench on Qwen-Image-Edit, and even distilled models. In all settings, HyCa delivers state-of-the-art performance.

## 2 RELATED WORK

Diffusion models (Sohl-Dickstein et al., 2015; Ho et al., 2020) have achieved strong image/video generation quality. Early U-Net backbones (Ronneberger et al., 2015) faced scaling limits that Diffusion Transformers (DiT) (Peebles & Xie, 2023b) alleviated, enabling rapid progress across modalities and resolutions (Chen et al., 2024b;a; Zheng et al., 2024; Yang et al., 2025). Nevertheless, the iterative nature of sampling remains a key inference bottleneck. Two complementary research lines thus emerge: (i) reducing the number of steps and (ii) reducing the cost per step. Beyond speed, a central challenge is maintaining stability and fidelity under aggressive acceleration, especially when feature dynamics are heterogeneous across dimensions and timesteps.

### 2.1 SAMPLING TIMESTEP REDUCTION

DDIM (Song et al., 2021) introduced deterministic few-step sampling that preserves perceptual quality. Higher-order ODE solvers (DPM-Solver and variants) (Lu et al., 2022a;b; Zheng et al., 2023) improve accuracy–cost trade-offs via multi-step/multi-stage discretizations with carefully controlled local truncation error. Rectified Flow (Liu et al., 2023) shortens transport paths, while distillation (Salimans & Ho, 2022) compresses long trajectories into compact generators. Consistency models (Song et al., 2023) enable few-step synthesis by learning a direct noise-to-clean mapping.

### 2.2 DENOISING NETWORK ACCELERATION

**Model Compression Acceleration.** Pruning (Fang et al., 2023; Zhu et al., 2024), quantization (Li et al., 2023b; Shang et al., 2023; Kim et al., 2025), distillation (Li et al., 2024), and token reduction (Bolya & Hoffman, 2023; Kim et al., 2024; Zhang et al., 2024; 2025; Cheng et al., 2025) reduce compute with limited runtime overhead. While effective, they typically require additional training and may degrade robustness under domain shifts if the compression is too aggressive.

**Feature Caching Acceleration.** Feature caching reuses activations to avoid redundant computation. Early U-Net methods (Li et al., 2023a; Ma et al., 2024) inspired DiT-specific designs: FasterCache (Lv et al., 2025), FORA (Selvaraju et al., 2024), $\Delta$-DiT (Chen et al., 2024c), TeaCache (Liu et al., 2024), and FoCa (Zheng et al., 2025). Dynamic updates (ToCa/DuCa) (Zou et al., 2024a;b), unified cache–prune pipelines (Sun et al., 2025), and region-adaptive sampling (Liu et al., 2025c) further improve efficiency. Among these advances, TaylorSeer (Liu et al., 2025a) exemplifies the *cache-then-forecast* paradigm by polynomial extrapolation from cached neighbors.

## 3 METHOD

### 3.1 PRELIMINARY

**Diffusion Models.** Diffusion models (Ho et al., 2020; Song et al., 2021) generate structured data by progressively refining random noise through a series of denoising steps. At each timestep $t$, the model predicts a conditional Gaussian distribution over $x_{t-1}$ given $x_t$, where both the mean and variance are parameterized. This generative process can be formulated as:

$$p_\theta(x_{t-1}|x_t) = \mathcal{N}\left(x_{t-1}; \frac{1}{\sqrt{\alpha_t}}\left(x_t - \frac{1-\alpha_t}{\sqrt{1-\bar{\alpha}_t}}\tau_\theta(x_t, t)\right), \beta_t \mathbf{I}\right), \tag{1}$$

where $\mathcal{N}$ denotes a normal distribution, $\alpha_t$ and $\beta_t$ are noise schedule parameters, and $\tau_\theta(x_t, t)$ denotes the model's estimate of the noise component. Sampling begins from a pure noise vector and proceeds by repeatedly drawing samples from these intermediate distributions until a clean image is produced.

**Diffusion Transformer Architecture.** The Diffusion Transformer (DiT) (Peebles & Xie, 2023a) adopts a hierarchical design, expressed as a composition of modules $\mathcal{G} = g_1 \circ g_2 \circ \cdots \circ g_L$. Each module $g_l$ consists of a self-attention layer ($\mathcal{F}_{SA}^l$), a cross-attention layer ($\mathcal{F}_{CA}^l$), and a feedforward MLP ($\mathcal{F}_{MLP}^l$). These components are dynamically modulated across timesteps to accommodate the evolving noise levels during generation. The input $\mathbf{x}_t = \{x_i\}_{i=1}^{H \times W}$ is represented as a sequence of patch tokens. Each module includes a residual update of the form $\mathcal{F}(\mathbf{x}) = \mathbf{x} + \text{AdaLN} \circ f(\mathbf{x})$, where AdaLN (adaptive layer normalization) conditions the normalization parameters on the noise timestep, allowing for more effective denoising across varying noise scales.

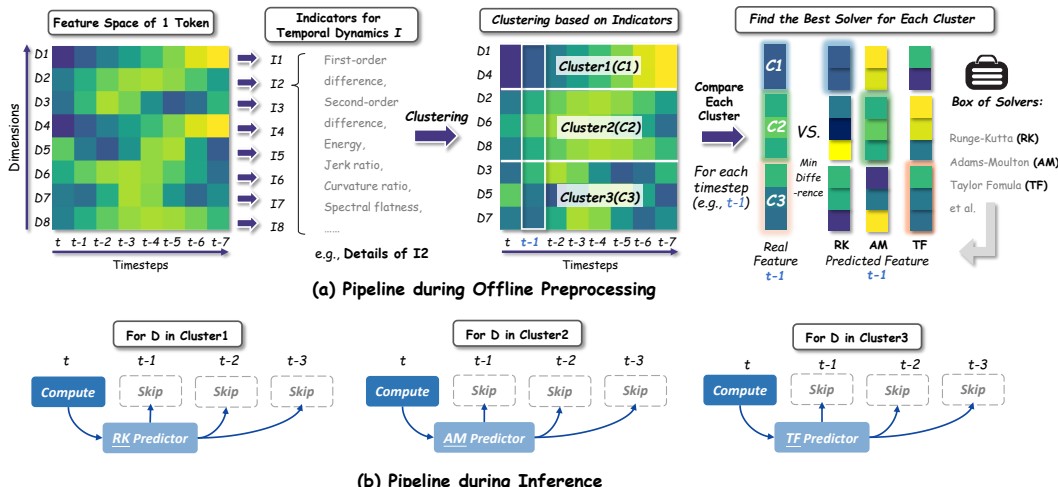

Figure 3: **HyCa Framework.** (a) Offline Preprocessing: feature dimensions are first analyzed and clustered with temporal indicators (e.g., differences, curvature). For each cluster, candidate solvers generate predicted features, then compared against real computed features; the solver with minimum error is then assigned to that cluster. (b) Inference: once assigned, each cluster consistently reuses its solver, enabling efficient prediction by skipping redundant computations while maintaining accuracy.

**Feature Caching.** Feature caching aims to reduce the cost of diffusion sampling by avoiding repeated computation of hidden features across timesteps. At each timestep $t$, the model produces hidden features $\mathcal{F}_t = \{\mathcal{F}_t^l\}_{l=1}^L$, and a caching function $\mathcal{C}(\mathcal{F}_A, k)$ estimates features $\tilde{\mathcal{F}}_k$ at a future timestep $k$ using cached features. A common strategy is to reuse features from the last computed step:

$$\tilde{\mathcal{F}}_k = \mathcal{C}(\mathcal{F}_t, k) := \mathcal{F}_t, \quad \forall k \in (t, t + n - 1], \tag{2}$$

which provides up to $(n-1)\times$ speedup. Recent methods improve reuse by forecasting future features, yet their reliance on a uniform prediction strategy across all dimensions often proves unstable in DiT's complex hidden feature space. In this work, we propose a hybrid approach that assigns suitable solvers for every dimension according to their dynamic behaviors.

## 3.2 FEATURE CACHING AS HYBRID ODE SOLVING

During reverse-time denoising in diffusion models, the hidden features evolve across timesteps. Let $x(\tau)$ be the latent variable at continuous time $\tau$, and let $\mathcal{F}(x(\tau))$ denote the hidden feature extracted from it. Since the generative model is differentiable and $x(\tau)$ follows a continuous reverse-time trajectory, the composite feature map $\tau \mapsto \mathcal{F}(x(\tau))$ is also differentiable. By the chain rule and the probability flow ODE governing $x(\tau)$, the feature dynamics satisfy:

$$\frac{d}{d\tau} \mathcal{F}(x(\tau)) = g_\theta\big(\mathcal{F}(x(\tau)), \tau\big), \tag{3}$$

where $g_\theta$ captures the implicit time-dependent vector field induced by the underlying network weights and structure. Although $g_\theta$ is not directly accessible, we can sample the trajectory $\{\mathcal{F}(x(\tau_k))\}$ on a discrete timestep grid, enabling numerical integration using only cached feature values. This perspective naturally casts feature caching as a numerical ODE solving problem. Instead of performing full forward computation at every (discrete) sampler step, we aim to solve the next feature value using prior ones:

$$\hat{\mathcal{F}}_{t+1} \approx \text{Solver}(\mathcal{F}_t, \mathcal{F}_{t-1}, \dots), \tag{4}$$

where the solver is applied locally to the residual output of each transformer block at skip steps. To accommodate diverse local feature dynamics, from smooth near-linear segments to rapidly varying regions, we adopt a hybrid solver strategy. Concretely, we define a predictor pool $\mathcal{S}$ that includes both explicit and implicit numerical solvers with different stability and accuracy properties, including Runge–Kutta (RK), Adams–Bashforth (AB), Taylor Formula (TF), Backward Differentiation Formula (BDF) and Adams–Moulton (AM). This diverse solver set enables HyCa to assign methods tailored to local feature dynamics. *Please refer to A.2.2 and A.6 for detailed implementation.*

## 3.3 HyCa Framework

Building on this foundation, **HyCa** is designed as a feature caching framework that models hidden dynamics as a mixture of ODEs and automatically assigns the most suitable solver to each cluster through a one-time optimization procedure. HyCa begins by analyzing the temporal dynamic behavior of each feature dimension. During a probe pass on a single prompt at the first few timesteps, we extract a descriptor vector $\phi_d \in \mathbb{R}^k$ for each feature dimension $d \in \{1, \ldots, D\}$, capturing dynamic indicators such as Jerk ratio and curvature ratio. Then, we apply $k$-means clustering to obtain a partition $\{c(d)\}$, where each dimension $d$ is assigned to a cluster $c \in \{1, \ldots, C\}$. These cluster assignments remain stable across prompts, timesteps and resolutions, thus reused throughout inference.

The resulting clusters represent groups of dimensions that share similar temporal behaviors, enabling solver assignments to be conducted independently for each cluster. Given a solver pool $\mathcal{S}$, HyCa selects the optimal solver $s_c^\star \in \mathcal{S}$ for each cluster $c$ by minimizing the average next-step prediction error across all dimensions in that cluster:

$$\min_{\{s_c \in \mathcal{S}\}_{c=1}^{C}} \sum_{c=1}^{C} \left[ \frac{1}{|c|} \sum_{d \in c} \left\| \hat{\mathcal{F}}_{t+1}^{(s_c, d)} - \mathcal{F}_{t+1}^{(d)} \right\|_2^2 \right], \tag{5}$$

where $\hat{\mathcal{F}}_{t+1}^{(s_c, d)}$ denotes the predicted feature for dimension $d$ at timestep $t + 1$ using solver $s_c$. This formulation enables per-cluster solver selection via a one-time probing pass, ensuring that HyCa combines efficiency with the adaptability of hybrid solvers.

## 4 Experiments

### 4.1 Experiment Settings

**Model Configurations.** We conduct experiments on four representative diffusion-based models: the text-to-image models FLUX.1-dev (Labs, 2024) and Qwen-Image (Wu et al., 2025), the text-to-video model HunyuanVideo (Kong et al., 2024), and the image editing model Qwen-Image-Edit (Wu et al., 2025). To further assess compatibility with model compression techniques, we also evaluate our method on distilled models: FLUX.1-schnell and Qwen-Image-Lightning. All models are evaluated under official or recommended configurations on standard public checkpoints.

**Evaluation and Metrics.** For text-to-image generation, we follow the DrawBench (Saharia et al., 2022) protocol and evaluate all models on a fixed set of 200 prompts. We evaluate images using ImageReward (Xu et al., 2023) for photorealism, CLIP Score (Hessel et al., 2021) for text–image alignment, and PSNR, SSIM, LPIPS for fidelity. For text-to-video generation, we evaluate Hunyuan-Video on VBench (Huang et al., 2023), which provides multi-dimensional human-aligned assessments of motion quality, visual appearance, and semantic consistency. For image editing tasks, we use GEdit-Bench (Liu et al., 2025b) to evaluate model performance across a diverse set of edit types and prompts. Unless otherwise specified, all evaluations are conducted using fixed random seeds and default inference settings. *Additional implementation details please refer to A.1.*

Table 1: **Quantitative comparison of text-to-image generation** on Qwen-Image.

| Method | Acceleration | | | | Quality Metrics | | Perceptual Metrics | | |
|---|---|---|---|---|---|---|---|---|---|
| | Latency(s) ↓ | Speed ↑ | FLOPs(T) ↓ | Speed ↑ | Image Reward ↑ | CLIP ↑ | PSNR ↑ | SSIM ↑ | LPIPS ↓ |
| **Original: 50 steps** | 74.91 | 1.00× | 12917.56 | 1.00× | 1.2547 (+0.000%) | 35.51 | ∞ | 1.000 | 0.000 |
| 50% steps | 37.73 | 1.99× | 6458.78 | 2.00× | 1.2048 (-3.979%) | 35.31 | 30.85 | 0.798 | 0.249 |
| 20% steps | 15.31 | 4.89× | 2583.51 | 5.00× | 0.9234 (-26.42%) | 35.17 | 28.52 | 0.627 | 0.504 |
| **TaylorSeer**($\mathcal{N} = 3$) | 36.90 | 2.03× | 4646.60 | 2.78× | 1.0685 (-14.83%) | 34.76 | 28.29 | 0.504 | 0.628 |
| **HyCa**($\mathcal{N} = 3$) | **35.33** | **2.12×** | **4646.60** | **2.78×** | **1.2363** (-1.465%) | **34.97** | **30.42** | **0.763** | **0.247** |
| **FORA**($\mathcal{N} = 5$) | 21.71 | 3.45× | 2585.46 | 5.00× | 0.7767 (-38.10%) | 34.47 | 24.55 | 0.553 | 0.556 |
| **ToCa**($\mathcal{N} = 8$) | 60.62 | 1.24× | 2991.34 | 4.32× | 0.9673 (-22.87%) | 34.83 | 29.00 | 0.643 | 0.417 |
| **DuCa**($\mathcal{N}=9$) | 24.83 | 3.02× | 2958.13 | 4.37× | 0.8213 (-34.53%) | 34.69 | 28.42 | 0.582 | 0.531 |
| **TaylorSeer**($\mathcal{N} = 6$) | 24.61 | 3.04× | 2585.46 | 5.00× | 0.9483 (-24.41%) | 34.76 | 28.29 | 0.504 | 0.628 |
| **HyCa**($\mathcal{N} = 6$) | **21.58** | **3.47×** | **2584.46** | **5.00×** | **1.1939** (-4.848%) | **34.87** | **29.65** | **0.709** | **0.320** |
| **FORA**($\mathcal{N} = 6$) | 17.89 | 4.19× | 2323.30 | 5.56× | 0.4781 (-61.91%) | 28.50 | 28.38 | 0.546 | 0.597 |
| **ToCa**($\mathcal{N} = 12$) | 52.72 | 1.42× | 2406.20 | 5.37× | 0.5593 (-55.42%) | 33.92 | 28.72 | 0.589 | 0.519 |
| **DuCa**($\mathcal{N}=12$) | 21.82 | 3.43× | 2171.56 | 5.95× | 0.5225 (-58.34%) | 33.97 | 28.37 | 0.576 | 0.593 |
| **TaylorSeer**($\mathcal{N} = 7$) | 21.88 | 4.32× | 2323.30 | 5.56× | 0.9113 (-27.39%) | 34.30 | 28.20 | 0.481 | 0.652 |
| **HyCa**($\mathcal{N} = 8$) | **13.92** | **5.38×** | **2066.81** | **6.25×** | **1.0811** (-13.84%) | **34.75** | **28.89** | **0.600** | **0.433** |

Table 2: **Quantitative comparison of text-to-image generation** on FLUX.1-dev.

| Method FLUX.1 | Efficient Attention | Acceleration | | | | Image Reward ↑ | CLIP Score ↑ |
|---|---|---|---|---|---|---|---|
| | | Latency(s) ↓ | Speed ↑ | FLOPs(T) ↓ | Speed ↑ | | |
| **[dev]: 50 steps** | ✔ | 25.82 | 1.00× | 3719.50 | 1.00× | 0.9898 (+0.000%) | 32.404 (+0.000%) |
| 60% **steps** | ✔ | 16.70 | 1.55× | 2231.70 | 1.67× | 0.9663 (-2.371%) | 32.312 (-0.283%) |
| $\Delta$-DiT ($\mathcal{N}=2$) | ✔ | 17.80 | 1.45× | 2480.01 | 1.50× | 0.9444 (-4.594%) | 32.273 (-0.404%) |
| $\Delta$-DiT ($\mathcal{N}=3$) | ✔ | 13.02 | 1.98× | 1686.76 | 2.21× | 0.8721 (-11.90%) | 32.102 (-0.933%) |
| **DBcache** | ✔ | 16.88 | 1.53× | 2384.29 | 1.56× | 1.0069 (+1.725%) | 32.530 (+0.389%) |
| **TaylorSeer** ($\mathcal{N}=3, O=2$) | ✔ | 9.89 | 2.61× | 1320.07 | 2.82× | 0.9989 (+0.919%) | 32.413 (+0.027%) |
| **FoCa** ($\mathcal{N}=3$) | ✔ | 9.28 | 2.78× | 1327.21 | 2.80× | 0.9890 (-0.081%) | 32.577 (+0.533%) |
| **HyCa** ($\mathcal{N}=4$) | ✔ | **8.09** | **3.19×** | **967.91** | **3.84×** | **1.0182 (+2.865%)** | **32.671 (+0.822%)** |
| 34% **steps** | ✔ | 9.07 | 2.85× | 1264.63 | 3.13× | 0.9453 (-4.498%) | 32.114 (-0.893%) |
| **Chipmunk** | ✔ | 12.72 | 2.02× | 1505.87 | 2.47× | 0.9936 (+0.384%) | 32.548 (+0.444%) |
| **FORA** ($\mathcal{N}=3$) | ✔ | 10.16 | 2.54× | 1320.07 | 2.82× | 0.9776 (-1.232%) | 32.266 (-0.425%) |
| **ToCa** ($\mathcal{N}=6$) | ✘ | 13.16 | 1.96× | 924.30 | 4.02× | 0.9802 (-0.968%) | 32.083 (-0.990%) |
| **DuCa** ($\mathcal{N}=5$) | ✔ | 8.18 | 3.15× | 978.76 | 3.80× | 0.9955 (+0.576%) | 32.241 (-0.503%) |
| **TaylorSeer** ($\mathcal{N}=4, O=2$) | ✔ | 9.24 | 2.80× | 967.91 | 3.84× | 0.9857 (-0.414%) | 32.413 (+0.027%) |
| **FoCa** ($\mathcal{N}=4$) | ✔ | 9.35 | 2.76× | 1050.70 | 3.54× | 0.9757 (-1.424%) | 32.538 (+0.414%) |
| **Clusca** ($\mathcal{N}=4, O=2, K=16$) | ✔ | 9.25 | 2.79× | 1045.58 | 3.56× | 0.9850 (-0.485%) | 32.441 (+0.114%) |
| **HyCa** ($\mathcal{N}=5$) | ✔ | **7.62** | **3.38×** | **893.54** | **4.16×** | **1.0066 (+1.700%)** | **32.693 (+0.890%)** |
| **FORA** ($\mathcal{N}=4$) | ✔ | 8.12 | 3.14× | 967.91 | 3.84× | 0.9730 (-1.695%) | 32.142 (-0.809%) |
| **ToCa** ($\mathcal{N}=8$) | ✘ | 11.36 | 2.27× | 784.54 | 4.74× | 0.9451 (-4.514%) | 31.993 (-1.271%) |
| **DuCa** ($\mathcal{N}=7$) | ✔ | **6.74** | **3.83×** | 760.14 | 4.89× | 0.9757 (-1.424%) | 32.066 (-1.046%) |
| **TeaCache** ($l=0.8$) | ✔ | 7.21 | 3.58× | 892.35 | 4.17× | 0.8683 (-12.28%) | 31.704 (-2.159%) |
| **TaylorSeer** ($\mathcal{N}=5, O=2$) | ✔ | 7.46 | 3.46× | 893.54 | 4.16× | 0.9768 (-1.314%) | 32.467 (+0.194%) |
| **FoCa** ($\mathcal{N}=6$) | ✔ | 7.54 | 3.42× | 745.39 | 4.99× | 0.9713 (-1.870%) | **32.922 (+1.600%)** |
| **Speca** ($\mathcal{N}_{max}=8, \mathcal{N}_{min}=2$) | ✔ | 7.42 | 3.48× | 791.38 | 4.70× | 0.9985 (+0.878%) | 32.277 (-0.391%) |
| **Clusca** ($\mathcal{N}=5, O=1, K=16$) | ✔ | 7.05 | 3.66× | 897.03 | 4.14× | 0.9718 (-1.818%) | 32.319 (-0.262%) |
| **HyCa** ($\mathcal{N}=6$) | ✔ | 6.81 | 3.79× | **744.81** | **5.00×** | **1.0014 (+1.163%)** | **32.483 (+0.244%)** |
| **FORA** ($\mathcal{N}=6$) | ✔ | 8.17 | 3.16× | 744.80 | 4.99× | 0.7760 (-21.62%) | 31.742 (-2.043%) |
| **ToCa** ($\mathcal{N}=10$) | ✘ | 7.93 | 3.25× | 714.66 | 5.20× | 0.7155 (-27.70%) | 31.808 (-1.839%) |
| **DuCa** ($\mathcal{N}=9$) | ✔ | 7.27 | 3.55× | 690.25 | 5.39× | 0.8382 (-15.33%) | 31.759 (-1.993%) |
| **TeaCache** ($l=1$) | ✔ | 8.19 | 3.19× | 743.63 | 5.01× | 0.8379 (-15.36%) | 31.877 (-1.627%) |
| **TaylorSeer** ($\mathcal{N}=7, O=2$) | ✔ | 6.77 | 3.81× | 671.39 | 5.54× | 0.9698 (-2.020%) | 32.128 (-0.851%) |
| **Clusca** ($\mathcal{N}=6, O=1, K=16$) | ✔ | 7.13 | 3.62× | 748.48 | 4.97× | 0.9704 (-1.956%) | 32.217 (-0.577%) |
| **HyCa** ($\mathcal{N}=7$) | ✔ | **6.43** | **4.01×** | **670.44** | **5.55×** | **0.9895 (-0.030%)** | **32.520 (+0.358%)** |

## 4.2 RESULTS ON TEXT-TO-IMAGE GENERATION

As shown in Table 1 HyCa achieves the best overall trade-off on **Qwen-Image** across all compression levels. At $\mathcal{N}=3$, it matches TaylorSeer in speed (2.78×) but yields higher quality (ImageReward **1.2363** vs. 1.0685, and highest PSNR **30.42**). At $\mathcal{N}=6$, HyCa remains strong (**1.1939**, **29.65**), outperforming TaylorSeer (0.9483), ToCa (0.9673), and FORA (0.7767). Even at $\mathcal{N}=8$, it sustains good visual quality (**1.0811** at 6.25×), while others drop sharply (ToCa 0.6326, FORA 0.4781). These results highlight HyCa's robustness under high acceleration while preserving visual fidelity.

On Table 2, HyCa consistently achieves the best speed–quality trade-off on **FLUX.1-dev**. At moderate acceleration ($\mathcal{N}=5$), it reaches ImageReward of **1.0066** with 4.16× FLOPs reduction, surpassing TaylorSeer (0.9857 at 3.84×) and DuCa (0.9955 at 3.80×). Even under aggressive settings, it maintains superior quality: **1.0014** at 5.00× and even **0.9895** at 5.55×, closely matching the original model (**0.9898**) while other baselines degrade (TeaCache 0.8683, ToCa 0.7155). Visual comparison on Fig. 4 further confirms its advantage in image quality under high compression.

Table 3: **Quantitative comparison of text-to-video generation** on HunyuanVideo.

| Method HunyuanVideo | Efficient Attention | Acceleration | | | | VBench ↑ Score(%) |
|---|---|---|---|---|---|---|
| | | Latency(s) ↓ | Speed ↑ | FLOPs(T) ↓ | Speed ↑ | |
| **Original: 50 steps** | ✔ | 145.00 | 1.00× | 29773.0 | 1.00× | 80.66 (+0.0%) |
| 22% **steps** | ✔ | 31.87 | 4.55× | 6550.1 | 4.55× | 78.74 (-2.4%) |
| **TeaCache**($l=0.4$) | ✔ | 30.49 | 4.76× | 6550.1 | 4.55× | 79.36 (-1.6%) |
| **FORA**($N=5$) | ✔ | 34.39 | 4.22× | 5960.4 | 5.00× | 78.83 (-2.3%) |
| **ToCa** ($\mathcal{N}=5, R=90\%$) | ✘ | 38.52 | 3.76× | 7006.2 | 4.25× | 78.86 (-2.2%) |
| **DuCa** ($\mathcal{N}=5, R=90\%$) | ✔ | 31.69 | 4.58× | 6483.2 | 4.48× | 78.72 (-2.4%) |
| **TaylorSeer** ($\mathcal{N}=5, O=1$) | ✔ | 34.84 | 4.16× | 5960.4 | 5.00× | 79.93 (-0.9%) |
| **Speca** ($\mathcal{N}_{max}=8, \mathcal{N}_{min}=2$) | ✔ | 34.58 | 4.19× | 5692.7 | 5.23× | 79.98 (-0.8%) |
| **Clusca** ($\mathcal{N}=5, O=1, K=16$) | ✔ | 35.37 | 4.10× | 5373.0 | 5.54× | 79.96 (-0.9%) |
| **FoCa** ($\mathcal{N}=5$) | ✔ | 34.52 | 4.20× | 5966.5 | 4.99× | 79.96 (-0.8%) |
| **HyCa** ($\mathcal{N}=6$) | ✔ | **28.48** | **5.09×** | **5359.1** | **5.56×** | **80.25 (-0.5%)** |

| Original | HyCa | TeaCache | FORA | ToCa | DuCa | TaylorSeer |
|:---:|:---:|:---:|:---:|:---:|:---:|:---:|
| ×1.00 | ×5.55 | ×5.55 | ×5.55 | ×5.20 | ×5.20 | ×5.55 |

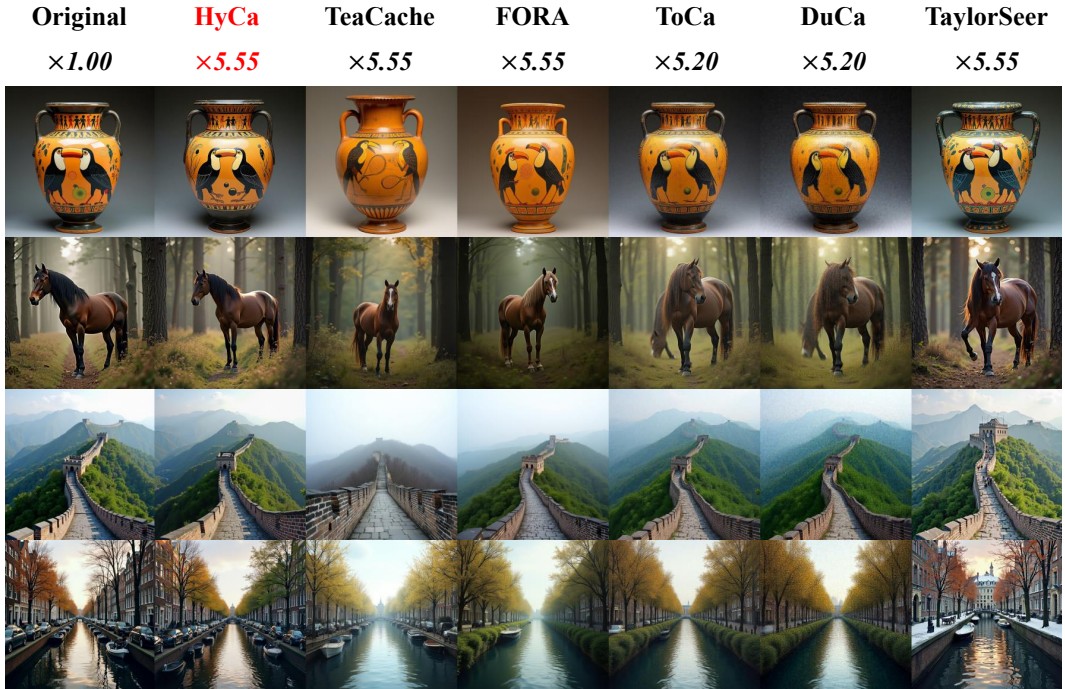

Figure 4: Visual comparison of 5.5× accelerated FLUX.

## 4.3 RESULTS ON TEXT-TO-VIDEO GENERATION

As shown in Table 3, our method delivers the best performance on **Hunyuan Video**. With $\mathcal{N}=6$, it achieves the highest acceleration (**5.56×** FLOPs reduction) while maintaining a strong VBench score (**80.25**), marginly lower than the original (**80.66**) at full 50-step inference. In contrast, TaylorSeer reaches only 4.16× with 79.93, TeaCache drops to 79.36, and DuCa/ToCa degrade further. This demonstrates a superior speed–quality trade-off and strong generalization to video generation.

Table 4: **Quantitative comparison of image editing** on Qwen-Image-Edit.

| Method | Acceleration | | | | GEdit-CN (Full) | | | GEdit-EN (Full) | | |
|---|---|---|---|---|---|---|---|---|---|---|
| | Latency(s) ↓ | Speed ↑ | FLOPs(T) ↓ | Speed ↑ | SC ↑ | PQ ↑ | OS ↑ | SC ↑ | PQ ↑ | OS ↑ |
| **Original: 50 steps** | 284.51 | 1.00× | 28190.88 | 1.00× | 7.68 | 7.51 | 7.41 | 7.82 | 7.54 | 7.54 |
| 50% steps | 143.29 | 1.99× | 14095.44 | 2.00× | 7.70 | 7.53 | 7.44 | 7.77 | 7.52 | 7.47 |
| 20% steps | 58.45 | 4.87× | 5638.18 | 5.00× | 7.65 | 7.42 | 7.35 | 7.73 | 7.46 | 7.44 |
| **FORA**($\mathcal{N}=5$) | 63.15 | 4.51× | 5643.13 | 5.00× | 7.60 | 7.31 | 7.25 | 7.62 | 7.34 | 7.28 |
| **DuCa** ($\mathcal{N}=6, R=90\%$) | 70.95 | 4.01× | 5897.67 | 4.78× | 7.63 | 7.44 | 7.44 | 7.68 | 7.42 | 7.39 |
| **TaylorSeer**($\mathcal{N}=6$) | 65.66 | 4.33× | 5643.13 | 5.00× | 7.25 | 7.09 | 6.92 | 7.26 | 7.14 | 6.89 |
| **HyCa** ($\mathcal{N}=6$) | **62.89** | **4.52×** | **5642.24** | **5.00×** | **7.76** | **7.49** | **7.50** | **7.77** | **7.47** | **7.45** |
| **FORA**($\mathcal{N}=7$) | 52.20 | 5.45× | 4515.74 | 6.24× | 7.42 | 7.13 | 7.06 | 7.43 | 7.19 | 7.06 |
| **DuCa** ($\mathcal{N}=10, R=95\%$) | 59.81 | 4.76× | 5158.45 | 5.46× | 7.50 | 5.75 | 6.39 | 7.52 | 5.77 | 6.41 |
| **TaylorSeer**($\mathcal{N}=8$) | 53.92 | 5.28× | 4515.74 | 6.24× | 6.61 | 6.65 | 6.31 | 6.67 | 6.63 | 6.31 |
| **HyCa** ($\mathcal{N}=8$) | **51.09** | **5.57×** | **4514.48** | **6.24×** | **7.74** | **7.41** | **7.44** | **7.80** | **7.36** | **7.42** |

- SC denotes Semantic Consistency on Gedit Bench, PQ denotes Perceptual Quality, OS denotes the Overall Score.

## 4.4 RESULTS ON IMAGE EDITING

Our method also performs strongly on **Qwen-Image-Edit**, as shown in Table 4. At $\mathcal{N}=6$, it achieves the best overall scores (**7.50** CN / **7.45** EN), surpassing TaylorSeer (6.92 / 6.89), FORA (7.25 / 7.28), and even the original model's (7.41 CN). At $\mathcal{N}=8$, it remains leading (**7.44** CN / **7.42** EN) even exceeding the original model, while other baselines drop sharply (TaylorSeer 6.31 / 6.31). Visual comparisons in Fig. 5 further confirm HyCa's superiority across diverse editing tasks.

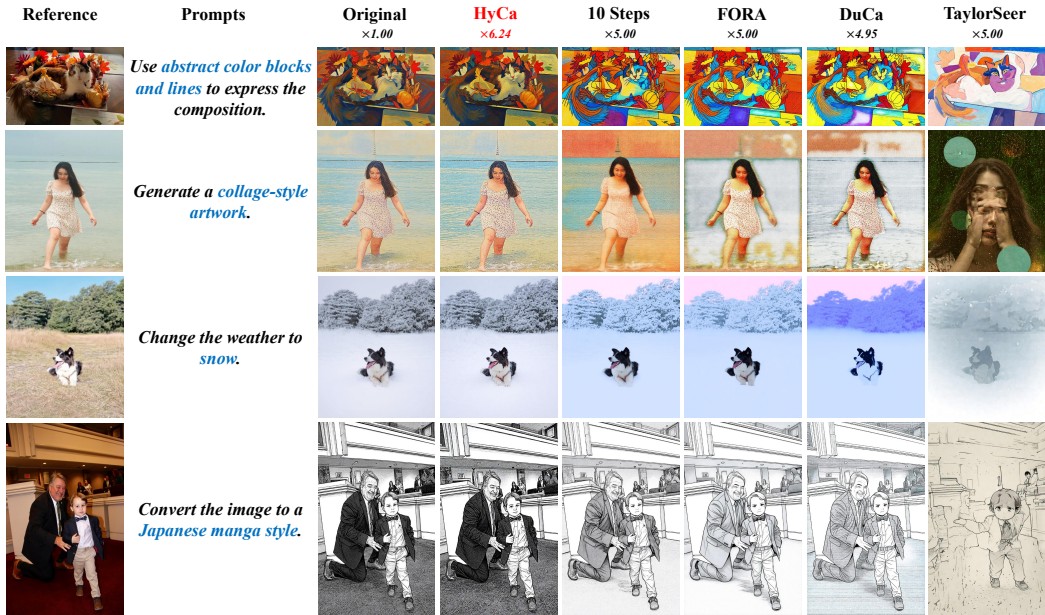

Figure 5: Visual comparison of different caching method on Qwen-Image-Edit.

## 4.5 RESULTS ON DISTILLED MODELS

To examine compatibility with distillation, we evaluate HyCa on **FLUX.1-schnell** and **Qwen-Image-Lightning**, as shown in Table 5. On FLUX.1-schnell, HyCa cuts latency from 2.34s to 1.16s for the 4-step distilled model (**24.4**× over the original 50-step model) while improving quality: ImageReward increased to **0.9592** and excellent perceptual metrics (PSNR 34.37, SSIM 0.928, LPIPS 0.056). On Qwen-Image-Lightning, it reduces latency from 13.35s(8-step distilled baseline) to 6.68s (**12.2**×) with quality largely preserved (ImageReward **1.2201**, CLIP 35.07) and perceptual metrics maintained (PSNR 30.97, SSIM 0.754, LPIPS 0.189). These results confirm that HyCa complements distillation, delivering extreme speedups with equal or even better quality.

Table 5: **Quantitative comparison of Distilled Model** on Flux and Qwen-Image.

| Method | Acceleration | | | | Quality Metrics | | Perceptual Metrics | | |
|---|---|---|---|---|---|---|---|---|---|
| | Latency(s)↓ | Speed ↑ | FLOPs(T)↓ | Speed ↑ | ImageReward↑ | CLIP↑ | PSNR↑ | SSIM↑ | LPIPS↓ |
| **FLUX.1[dev]: 50 steps** | 25.82 | 1.00× | 3719.50 | 1.00× | 0.9898 (+8.380%) | 32.40 | - | - | - |
| **FLUX.1[schnell]: 4 steps** | 2.34 | 11.03× | 297.60 | 12.50× | 0.9133 (+0.000%) | 33.85 | ∞ | 1.000 | 0.000 |
| **TaylorSeer ($\mathcal{N}=2$): 4 steps** | 1.58 | 16.34× | 209.70 | 17.74× | 0.9191 (+0.636%) | 33.76 | 29.13 | 0.746 | 0.249 |
| **TeaCache ($l=0.6$): 4 steps** | 1.26 | 20.49× | 163.78 | 22.71× | 0.9023 (-1.210%) | 33.87 | 28.01 | 0.379 | 0.734 |
| **HyCa ($\mathcal{N}=2$): 4 steps** | **1.16** | **22.25×** | **152.32** | **24.42×** | **0.9592 (+5.029%)** | **34.18** | **34.37** | **0.928** | **0.056** |
| **Qwen-Image: 50 steps** | 74.91 | 1.00× | 12917.56 | 1.00× | 1.2547 (+0.232%) | 35.51 | - | - | - |
| **Qwen-Image-Lightning: 8 steps** | 13.35 | 5.61× | 2113.67 | 6.11× | 1.2518 (+0.000%) | 35.32 | ∞ | 1.000 | 0.000 |
| **TaylorSeer ($\mathcal{N}=2$): 8 steps** | 8.11 | 9.24× | 1320.81 | 9.78× | 1.0418 (-16.79%) | 34.44 | 29.49 | 0.620 | 0.377 |
| **TaylorSeer ($\mathcal{N}=3$): 8 steps** | 6.78 | 11.07× | 1057.08 | 12.22× | 0.2644 (-78.89%) | 30.55 | 27.98 | 0.300 | 0.672 |
| **HyCa ($\mathcal{N}=2$): 8 steps** | 8.20 | 9.13× | 1320.81 | 9.78× | **1.2478 (-0.320%)** | **35.27** | 32.52 | **0.837** | 0.119 |
| **HyCa ($\mathcal{N}=3$): 8 steps** | **6.68** | **11.21×** | **1057.08** | **12.22×** | 1.2201 (-2.542%) | 35.07 | 30.97 | 0.754 | 0.189 |

- The PSNR, SSIM, and LPIPS of HyCa are computed with respect to the outputs of the corresponding distilled baseline models.

## 5 DISCUSSION

**Ablation Study.** We conduct our ablation study on FLUX, as shown in Fig. 7 (c)(d), HyCa consistently achieves higher ImageReward and lower prediction error than any individual solver baselines from our solver pool under the same conditions. These results confirm that HyCa benefits from combining diverse solvers rather than relying on a single integration strategy.

**Why Dimension-Wise Assignment?** A central design in HyCa is to assign caching strategies at feature-dimension level rather than token level primarily due to its better stability, as shown in Fig 6, clustering results in feature space remain nearly invariant once identified. Thus, solver assignments can be reused extensively, reducing both computational and data requirements. On the contrary, token-wise assignment varies with prompt and resolution, requiring frequent re-selection during

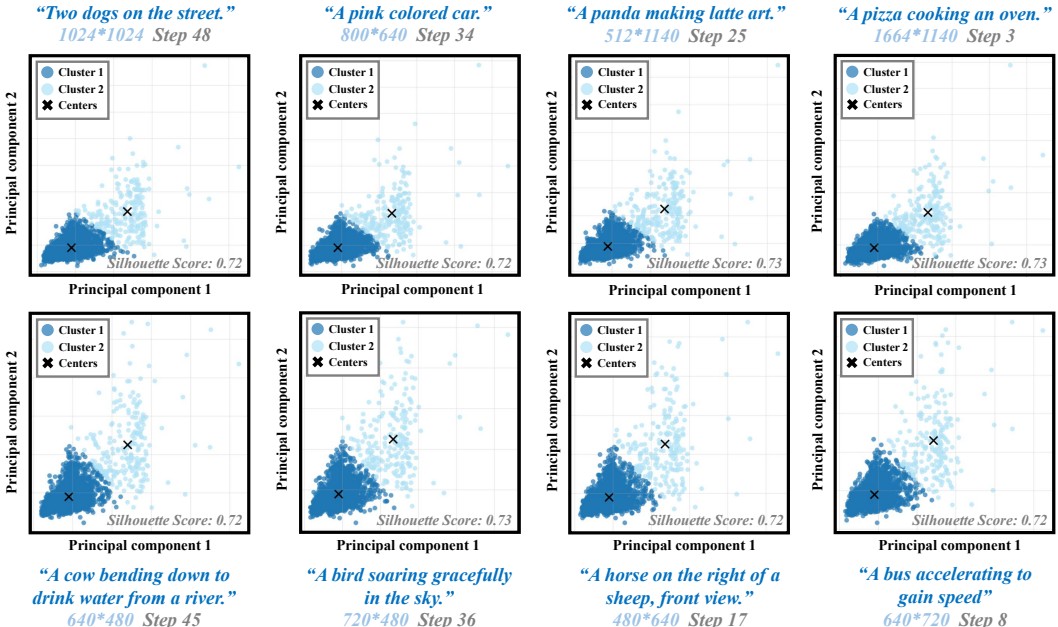

Figure 6: Top row: Clustering results from FLUX; Bottom row: Clustering results from Hunyuan Video. The clustering assignments remain highly consistent across various prompts, resolutions and timesteps, suggesting stable and robust geometric structure in the feature space.

inference. Empirically, Fig. 7 (a)(b) confirms that our dimension-wise assignment outperforms both token-wise (ToCa, DuCa) and one-size-fits-all (FORA, TaylorSeer) baselines.

**Compatibility with Distillation.** Feature caching is traditionally difficult to adapt to distilled models, as distillation drastically reduces sampling steps (e.g., from 50 to 4 or 8), making feature trajectories more discrete and oscillatory. Prior caching methods rely heavily on smooth temporal evolution and thus fail in this setting. In contrast, HyCa remains effective: its solver pool includes implicit methods suited for discrete or oscillatory dynamics, and solvers are assigned per cluster for each model. This flexibility makes HyCa fully compatible with distillation, achieving substantial acceleration while preserving generation quality.

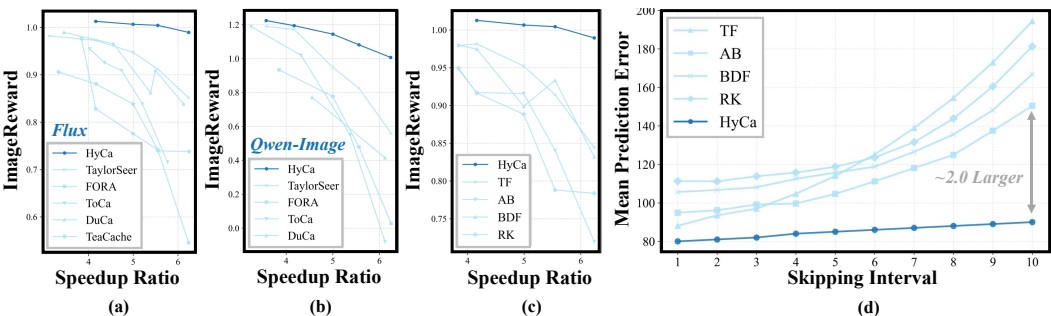

Figure 7: **Overall and ablation results of HyCa.** (a–b) HyCa consistently outperforms token-wise (ToCa, DuCa) and one-size-fits-all (FORA, Taylorseer) baselines on FLUX and Qwen-Image. (c–d) Ablation on FLUX shows HyCa surpasses all single-solver baselines in the solver pool, maintaining lower error and better quality. Confirming that HyCa benefits from combining diverse solvers rather than relying on a single integration strategy.

**Compatibility with LoRA Fine-Tuning.** HyCa remains effective under LoRA fine-tuning. As LoRA only updates a small subset of parameters through low-rank adapters, most feature dynamics and their clustering structures should remain stable. We evaluated this on XLabs-AI Art LoRA and Anime LoRA of FLUX (XLabs-AI, 2024), as shown in Fig. 8, the clustering results before and after

fine-tuning remain consistent. We further quantify this similarity by computing the Adjusted Rand Index (ARI) between the LoRA cluster assignments and the original model's. In both cases, the mean ARI exceeds 0.8, indicating strong agreement and confirming that HyCa's clustering results are largely invariant under LoRA adaptation. Therefore, we believe that for models sharing the same backbone, switching between different LoRA does not require re-clustering or solver re-selection in most practical cases. Even if re-computation is needed in rare cases, HyCa's offline cost remains minimal (about 1s). This demonstrates HyCa's robustness and efficiency in real-world deployment scenarios where multiple LoRA adapters are used for personalization or domain adaptation.

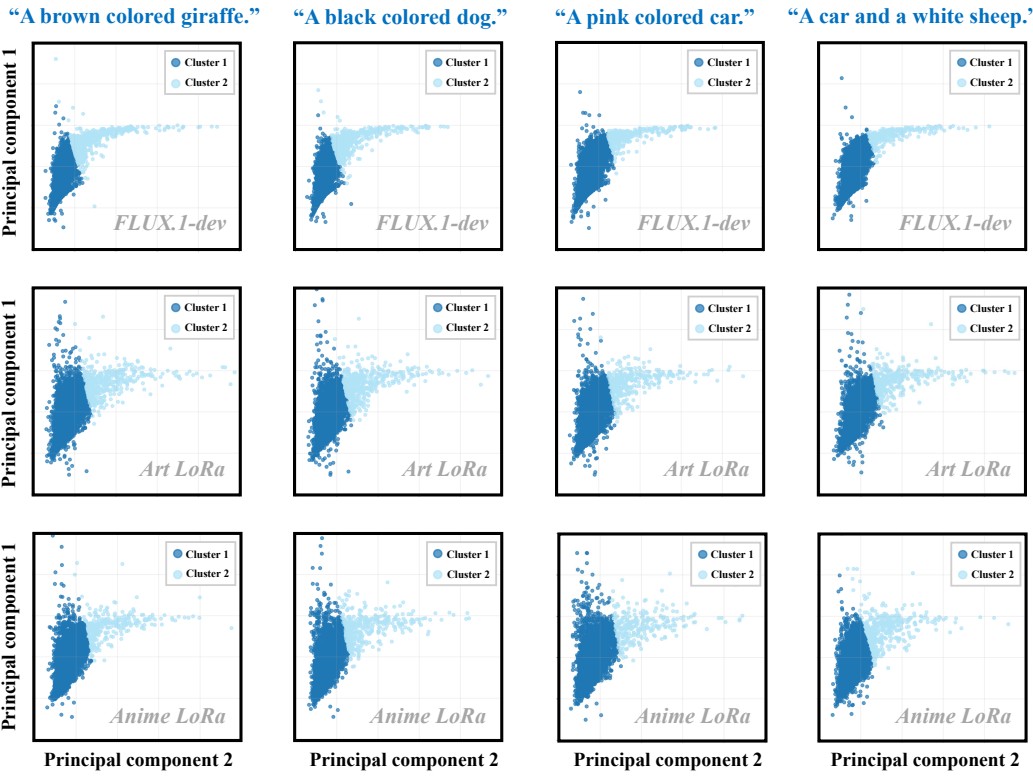

Figure 8: **Clustering consistency between original and LoRA-tuned models.** Each subplot visualizes the clustering results under different prompts for FLUX.1-dev (top row), and its two LoRA variants, the Art LoRA (middle row) and Anime LoRA (bottom row). Despite fine-tuning, the cluster boundaries remain highly consistent across prompts and models, indicating that HyCa's solver assignments remain stable and reusable even after LoRA adaptation.

## 6 CONCLUSION

In this work, we revisited feature caching in DiTs as hybrid solving ODEs. We revealed that the evolution of hidden features is not governed by a single, unified system, but rather by a mixture of ODEs, where different dimensions follow heterogeneous dynamics. Building on this insight, we introduce **HyCa**, a training-free framework that clusters feature dimensions and assigns tailored solvers from a hybrid solver pool. Analysis shows that the cluster structures in feature space are input-invariant, enabling *"One-Time Choosing"* and *"All-Time Solving"* with negligible overhead. Extensive experiments across text-to-image, text-to-video, and image-editing tasks demonstrate that HyCa achieves substantial acceleration without sacrificing quality, including **5.55**× acceleration on FLUX, **5.56**× on HunyuanVideo, **6.24**× on Qwen-Image, and even up to **22**× speedup on distilled variants. These results highlight its broad applicability to diverse architectures and the orthogonality to distillation. Looking forward, we envision extending this mixture-of-ODE perspective to other generative models and exploring learning-based caching strategies to further enhance efficiency.

## ACKNOWLEDGMENT

This project is sponsored by CCF-Tencent Rhino-Bird Funds.

## ETHICS STATEMENT

We acknowledge and adhere to the ICLR Code of Ethics in all aspects of this research. This work presents HyCa, a training-free acceleration framework for diffusion transformers that aims to improve computational efficiency without compromising generation quality. We address several ethical considerations relevant to our contribution:

**Responsible Use and Dual-Use Considerations.** While our method improves the efficiency of diffusion models used for creative and productive applications (text-to-image, text-to-video, image editing), we acknowledge that accelerated generative models could potentially be misused for creating misleading or harmful content such as deepfakes. However, our contribution is a technical acceleration method that is agnostic to content generation—it does not introduce new capabilities for harmful content creation but merely makes existing capabilities more efficient. The responsibility for ethical application remains with end users and deploying organizations.

**Data and Privacy.** Our method operates entirely on model activations and requires no additional training data or model modifications. We evaluate on standard public benchmarks (DrawBench, VBench, GEdit-Bench) without collecting or processing personal data. The caching mechanism operates on intermediate neural network representations and does not store or expose original input content, maintaining privacy during inference.

## REPRODUCIBILITY STATEMENT

To ensure full reproducibility of our results, we have made extensive efforts to document all implementation details and experimental configurations. Our code is provided as supplementary material, and will be released on Github.

All experimental protocols are thoroughly documented in A.1, including specific model configurations, evaluation datasets (DrawBench, VBench, GEdit-Bench), resolution settings, and random seed specifications (fixed seed=0 for all experiments). Our feature clustering methodology, including the descriptor vector construction using indicators such as Second-Order Change $d_2$, Curvature Ratios $\eta$, Jerk Ratios $\kappa$ and Spectral Flatness sf, can be found in A.2.1. The mathematical formulations for our hybrid ODE solver framework are provided in detail in A.2.2, with complete derivations of each numerical method in our predictor pool.

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

# A APPENDIX

## A.1 DETAILED EXPERIMENT SETTINGS

**Model Configurations and Evaluation Protocols.** Our experimental evaluation encompasses six distinct generative models across three primary tasks: text-to-image generation, text-to-video generation, and image editing. For text-to-image generation, we evaluate four different models: FLUX.1-dev, Qwen-Image, Flux-Schnell, and Qwen-Image-Lightning. Each model is assessed using standardized benchmarks and protocols to ensure fair comparisons. For text-to-video generation, we utilize the HunyuanVideo model, which is evaluated using the VBench framework to capture various aspects of video quality and coherence. Finally, for image editing tasks, we assess the Qwen-Image-Edit model using the GEdit-Bench benchmark, which provides a comprehensive suite of editing scenarios and evaluation metrics. Detailed configurations for each model, including resolution settings, prompt selections, and evaluation metrics, can be found in the subsequent sections.

### A.1.1 TEXT-TO-IMAGE GENERATION

**FLUX.1-dev.** We evaluate FLUX.1-dev using the standard DrawBench protocol, which provides a diverse set of 200 prompts spanning multiple categories including animals, colors, conflicting, and fine-grained details. All images are generated at a resolution of $1024 \times 1024$ pixels, maintaining consistency with the model's optimal operating parameters. We assess generation quality using two metrics: ImageReward (IR) for evaluating photorealism and overall image quality and CLIP Score for measuring text-image semantic alignment.

**Qwen-Image.** For the Qwen-Image model, we utilize the DrawBench dataset as well, which contains 200 prompts designed for evaluating image generation capabilities. The model generates images at a higher resolution of $1328 \times 1328$ pixels. We not only assess generation quality using ImageReward (IR) and CLIP Score, but also traditional fidelity metrics including Peak Signal-to-Noise Ratio (PSNR), Structural Similarity Index (SSIM), and Learned Perceptual Image Patch Similarity (LPIPS) for pixel-level and perceptual similarity assessment.

**FLUX.1-schnell.** FLUX.1-schnell, a step-distillation model from FLUX.1-dev, is designed for rapid image generation. It is evaluated using the standard DrawBench protocol with 200 prompts at $1024 \times 1024$ resolution. Despite its focus on speed optimization, we maintain the same evaluation standards, assessing both generation quality through ImageReward and CLIP Score, as well as fidelity through PSNR, SSIM, and LPIPS metrics. This allows us to quantify the trade-offs between generation speed and output quality.

**Qwen-Image-Lightning.** Qwen-Image-Lightning, the step-distilled variant of Qwen-Image is evaluated using 200 DrawBench prompts to maintain consistency with other text-to-image models, while operating at the model's native high resolution of $1328 \times 1328$ pixels. We also apply the same evaluation framework as above to assess the quality of the generated images.

### A.1.2 TEXT-TO-VIDEO GENERATION AND IMAGE EDITING

**HunyuanVideo.** For text-to-video generation, we evaluate HunyuanVideo using the comprehensive VBench framework, which provides multi-dimensional human-aligned assessments across 946 diverse prompts sourced from the VBench-full-info.json file. Videos are generated at $480 \times 640$ resolution with 65 frames each, providing substantial temporal content for thorough evaluation. VBench evaluates 18 critical aspects including but not limited to motion quality, visual appearance consistency, temporal coherence, semantic alignment with text prompts, and overall video quality through human-correlated metrics.

**Qwen-Image-Edit.** For image editing capabilities, we evaluate Qwen-Image-Edit using the GEdit-Bench framework, which contains 1212 carefully designed editing prompts covering 11 diverse edit types, including but not limited to background change, color alteration, material alteration, motion change, and pose manipulation. The benchmark provides comprehensive assessment across multiple editing scenarios without fixed resolution constraints, thus no specific resolution is enforced during evaluation. We utilize both the GEdit-CN and GEdit-EN subsets for evaluation, employing metrics such as Success Rate (SC) to measure the effectiveness of edits, Perceptual Quality (PQ) to assess visual fidelity, and Overall Score (OS) for a holistic evaluation of editing performance.

**Reproducibility and Consistency Measures.** All other settings, such as batch sizes, cfg_scale and model-specific parameters, are kept consistent with the original configurations recommended by the model developers. The standardized evaluation protocols ensure that observed performance differences reflect genuine model capabilities rather than experimental artifacts. *All implementation details and hyperparameter settings are carefully documented to facilitate reproduction of our experimental results.*

## A.2 DETAILED MATHEMATICAL FORMULATIONS

### A.2.1 CLUSTERING INDICATORS

In this section, we provide detailed mathematical definitions for the statistical feature indicators used in our clustering-based solver assignment strategy. These indicators capture various aspects of the temporal dynamics of feature dimensions, enabling effective clustering and solver selection.

**First-Order Difference.** The first-order change indicator measures the velocity of feature evolution, representing the Euclidean distance between current and previous time steps:

$$d_1 = \|f_k - f_{k-1}\| \tag{6}$$

Where $f_k$ denotes the feature value on timestep $k$. This metric captures the immediate rate of change in feature values and serves as a fundamental indicator of temporal dynamics.

**Second-Order Difference.** The second-order change indicator quantifies acceleration or curvature in the feature trajectory:

$$d_2 = \|f_k - 2f_{k-1} + f_{k-2}\| \tag{7}$$

This represents the Euclidean norm of the second-order difference, providing insights into the smoothness and stability of feature evolution patterns.

**Third-Order Difference.** The third-order change indicator measures jerk in the feature trajectory:

$$d_3 = \|f_k - 3f_{k-1} + 3f_{k-2} - f_{k-3}\| \tag{8}$$

This metric captures the Euclidean norm of the third-order difference, indicating abrupt changes in acceleration and helping identify highly dynamic or unstable features.

**Curvature Ratio.** The curvature ratio provides a normalized measure of curvature relative to velocity:

$$\eta = \frac{d_2}{d_1 + \epsilon} \tag{9}$$

where $\epsilon$ is a small constant to prevent division by zero. This ratio helps distinguish between features with different curvature characteristics while accounting for their overall rate of change.

**Jerk Ratio.** The jerk ratio measures the relative magnitude of jerk compared to curvature:

$$\kappa = \frac{d_3}{d_2 + \epsilon} \tag{10}$$

This indicator helps identify features with sudden acceleration changes and is particularly useful for detecting instabilities that require specialized numerical treatment.

**Direction Consistency.** The direction consistency indicator measures the consistency of directional changes in feature evolution:

$$\rho = \cos(\theta) = \frac{(f_k - f_{k-1}) \cdot (f_{k-1} - f_{k-2})}{\|f_k - f_{k-1}\|\|f_{k-1} - f_{k-2}\| + \epsilon} \tag{11}$$

This metric quantifies how consistently features change direction over time, with values close to 1 indicating consistent direction and values close to -1 indicating frequent direction reversals.

**Relative Change Rate.** The relative change rate normalizes the first-order change by the magnitude of the previous time step:

$$\gamma = \frac{d_1}{\|f_{k-1}\| + \epsilon} \tag{12}$$

This ratio provides scale-invariant information about the rate of change, enabling comparison across features with different magnitude scales.

**Energy.** The energy indicator represents the magnitude of the current feature vector:

$$e = \|f_k\| \tag{13}$$

This Euclidean norm provides information about the overall scale and importance of feature dimensions, helping to weight their contribution to clustering decisions.

**Low Frequency Ratio.** The low frequency ratio quantifies the proportion of power concentrated in low-frequency components:

$$\text{lfr} = \frac{\sum_{i=0}^{q-1} P_i}{\sum_{i=0}^{N-1} P_i + \epsilon} \tag{14}$$

where $P_i$ represents the power in the $i$-th DCT frequency component and $q$ corresponds to the first 20% of frequency components. This metric helps identify features with predominantly smooth, low-frequency evolution patterns.

**Spectral Flatness.** The spectral flatness measures the uniformity of the frequency distribution:

$$\text{sf} = \frac{\exp(\frac{1}{M} \sum_{j=1}^{M} \ln P_j)}{\frac{1}{M} \sum_{j=1}^{M} P_j + \epsilon} \tag{15}$$

where $M$ is the number of frequency components. This ratio of geometric mean to arithmetic mean of the DCT power spectrum indicates whether the feature evolution has a flat or peaked frequency profile, helping distinguish between noise-like and structured temporal patterns.

### A.2.2 PREDICTOR POOL FORMULATIONS

In this section, we provide the detailed mathematical formulations for the predictor pool $\mathcal{S}$ used in our HyCa framework. The predictor pool consists of both explicit and implicit numerical solvers, each with different stability and accuracy properties. For a general ordinary differential equation (ODE) of the form:

$$\frac{dy}{dt} = f(t, y), \quad y(t_0) = y_0 \tag{16}$$

we define the following numerical methods for advancing from time step $t_n$ to $t_{n+1} = t_n + h$, where $h$ is the step size.

**Runge-Kutta Methods (RK).** The family of Runge-Kutta methods provides high-order accuracy through multiple function evaluations per step. The general $s$-stage explicit Runge-Kutta method is formulated as:

$$k_i = f\left(t_n + c_i h, \; y_n + h \sum_{j=1}^{i-1} a_{ij} k_j\right), \quad i = 1, 2, \ldots, s \tag{17}$$

$$y_{n+1} = y_n + h \sum_{i=1}^{s} b_i k_i \tag{18}$$

where $a_{ij}$, $b_i$, and $c_i$ are the Butcher tableau coefficients that define the specific Runge-Kutta method. These coefficients satisfy consistency conditions:

$$c_i = \sum_{j=1}^{i-1} a_{ij}, \quad \sum_{i=1}^{s} b_i = 1 \tag{19}$$

The order of accuracy is determined by the number of order conditions satisfied by these coefficients.

**Adams-Bashforth Methods (AB).** These explicit multistep methods utilize information from previous time steps to achieve higher accuracy. The $k$-step Adams-Bashforth method is given by:

$$y_{n+1} = y_n + h \sum_{j=0}^{k-1} \beta_j f(t_{n-j}, y_{n-j}) \tag{20}$$

where $\beta_j$ are the Adams-Bashforth coefficients for the $k$-step method, which can be computed using the formula:

$$\beta_j = \frac{(-1)^j}{j!(k-1-j)!} \int_0^1 \prod_{\substack{i=0 \\ i \neq j}}^{k-1} (s+i)\, ds \tag{21}$$

**Taylor Series Extrapolation (TF).** This method extends the Taylor series expansion to higher orders for improved accuracy:

$$y_{n+1} = y_n + h f_n + \frac{h^2}{2!} f_n' + \frac{h^3}{3!} f_n'' + \cdots + \frac{h^p}{p!} f_n^{(p-1)} \tag{22}$$

where $f_n^{(k)}$ denotes the $k$-th derivative of $f$ evaluated at $(t_n, y_n)$. For practical implementation, we compute derivatives using automatic differentiation:

$$f_n' = \frac{\partial f}{\partial t} + \frac{\partial f}{\partial y} f \tag{23}$$

$$f_n'' = \frac{\partial^2 f}{\partial t^2} + 2\frac{\partial^2 f}{\partial t \partial y} f + \frac{\partial^2 f}{\partial y^2} f^2 + \frac{\partial f}{\partial y} f' \tag{24}$$

**Backward Differentiation Formula (BDF).** These implicit methods provide excellent stability properties for stiff equations. The $k$-step BDF method is formulated as:

$$\sum_{j=0}^k \alpha_j y_{n+1-j} = h\beta f(t_{n+1}, y_{n+1}) \tag{25}$$

where the coefficients $\alpha_j$ and $\beta$ are determined by the requirement that the method is exact for polynomials of degree $k$. The coefficients can be systematically computed through the differentiation of the interpolating polynomial through the points $(t_{n+1-j}, y_{n+1-j})$ for $j = 0, 1, \ldots, k$.

**Adams-Moulton Methods (AM).** These implicit multistep methods offer improved stability compared to their explicit counterparts:

$$y_{n+1} = y_n + h \sum_{j=-1}^{k-1} \beta_j f(t_{n+1+j}, y_{n+1+j}) \tag{26}$$

where the coefficients $\beta_j$ are obtained by integrating the interpolating polynomial through the points $(t_{n+1+j}, f_{n+1+j})$ for $j = -1, 0, \ldots, k-1$, resulting in a $k$-step implicit method with order $k+1$.

**Solver Selection Criteria.** Each method in the predictor pool $\mathcal{S}$ exhibits distinct stability and accuracy characteristics. Explicit methods including Runge-Kutta, Adams-Bashforth, and Taylor extrapolation offer computational efficiency but may suffer from stability constraints. In contrast, implicit methods such as BDF and Adams-Moulton provide superior stability for stiff problems at the cost of solving nonlinear equations. Higher-order methods achieve better accuracy but require more computational resources, while multistep methods utilize historical information efficiently but require special initialization procedures.

The adaptive selection mechanism in HyCa evaluates local feature dynamics and assigns the most appropriate solver from $\mathcal{S}$ based on stability requirements, accuracy demands, and computational constraints at each spatial location.

## A.3 VISUAL QUALITY COMPARISONS

### A.3.1 TEXT-TO-IMAGE GENERATION RESULTS

Figure 9 provides qualitative comparisons between our HyCa framework and baseline methods across different tasks. Each figure highlights the visual fidelity and detail preservation achieved by HyCa under high acceleration ratios.

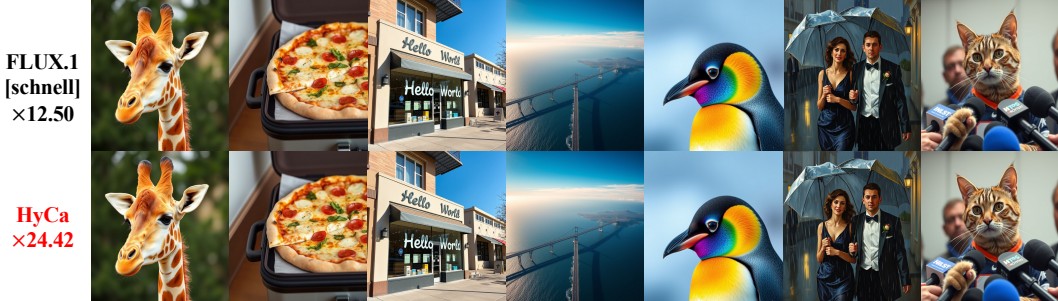

Figure 9: Comparison between HyCa and the original FLUX.1-Schnell model. HyCa achieves much higher acceleration ratio while producing images that are almost indistinguishable in quality from the originals.

### A.3.2 TEXT-TO-VIDEO GENERATION RESULTS

Figure 10 illustrates the superior temporal coherence and visual fidelity of videos generated by HyCa compared to TaylorSeer under high acceleration ratios. TaylorSeer exhibits noticeable artifacts and temporal inconsistencies, while HyCa maintains smooth motion and consistent details throughout the video sequence. Artifacts are highlighted with red boxes along with text clarifications.

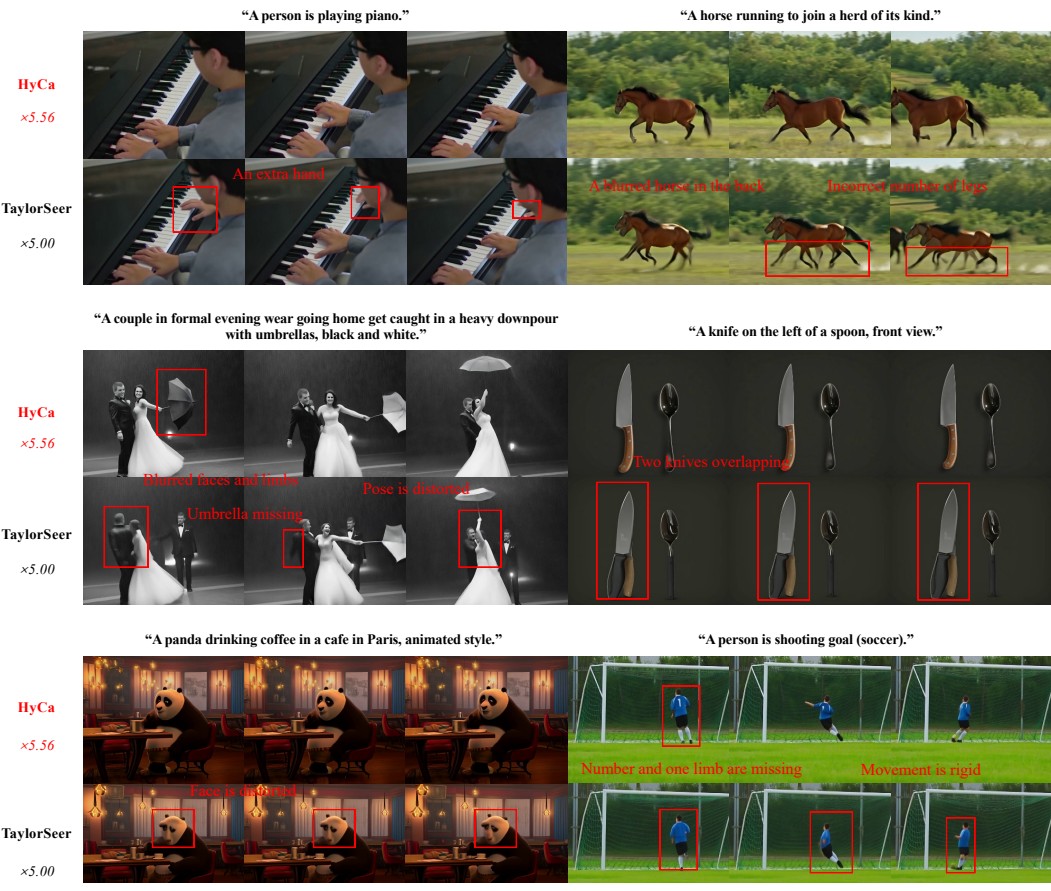

Figure 10: Comparison between HyCa and TaylorSeer on HunyuanVideo. HyCa maintains superior temporal coherence and visual fidelity under higher acceleration ratio, while TaylorSeer exhibits noticeable artifacts and temporal inconsistencies.

## A.4   CLUSTERING ANALYSIS AND SOLVER ASSIGNMENT

Our HyCa framework operates on a fundamental principle that feature dimensions with similar temporal dynamics should be grouped together and assigned the same numerical solver. This clustering-based approach ensures computational efficiency while maintaining prediction accuracy. The methodology follows a three-stage logical progression: (1) identifying feature dimensions with similar temporal behavior through clustering analysis, (2) characterizing these clusters using trajectory-based descriptors, and (3) assigning optimal solvers to each cluster based on their dynamic properties.

The clustering similarity is first established by analyzing the temporal evolution patterns of individual feature dimensions across multiple timesteps. Features exhibiting similar rates of change, curvature characteristics, and stability properties naturally form coherent clusters. Once these clusters are formed, we extract trajectory-based metrics such as second-order differences and curvature ratios to quantify the dynamic behavior within each cluster. These trajectory features serve as descriptors that capture the essential characteristics of feature evolution within each group. Finally, the solver selection process leverages these trajectory descriptors to match each cluster with the most suitable numerical method from our predictor pool, ensuring that explicit methods are assigned to stable, smooth regions while implicit methods handle stiff or rapidly varying dynamics.

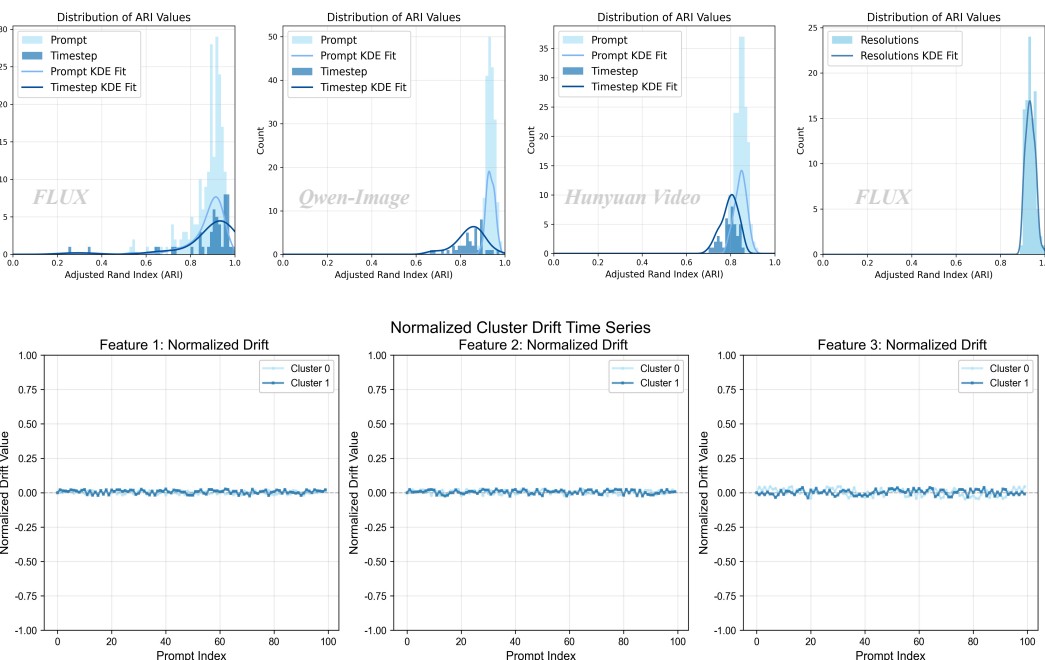

Figure 11: The first row shows, from left to right, ARI comparison plots of FLUX, Qwen-Image, and Hunyuan Video under different prompts and timesteps, with the last plot illustrating ARI comparisons of FLUX across different resolutions. It can be seen that ARI distributions exceed 0.8 in most cases, confirming stable and consistent cluster assignments across prompts, timesteps and resolutions. *An ARI above 0.8 indicates strong agreement and high clustering reliability.* The three plots in the second row depict the intra-cluster metric shifts between clusters formed by indicators from every two adjacent prompts of FLUX. The normalized value is obtained by dividing the inter-cluster shift of a given indicator by the average value of that indicator within the cluster. It can be seen that most values fluctuate between -0.02 and 0.02, indicating excellent consistency of the metrics.

## A.5   COMPATIBILITY WITH LORA FINE-TUNING

HyCa is also robust under LoRA fine-tuning. We evaluate HyCa on XLabs-AI Art LoRA and Anime LoRA of the FLUX.1-dev model and analyze whether clustering and solver assignments need to

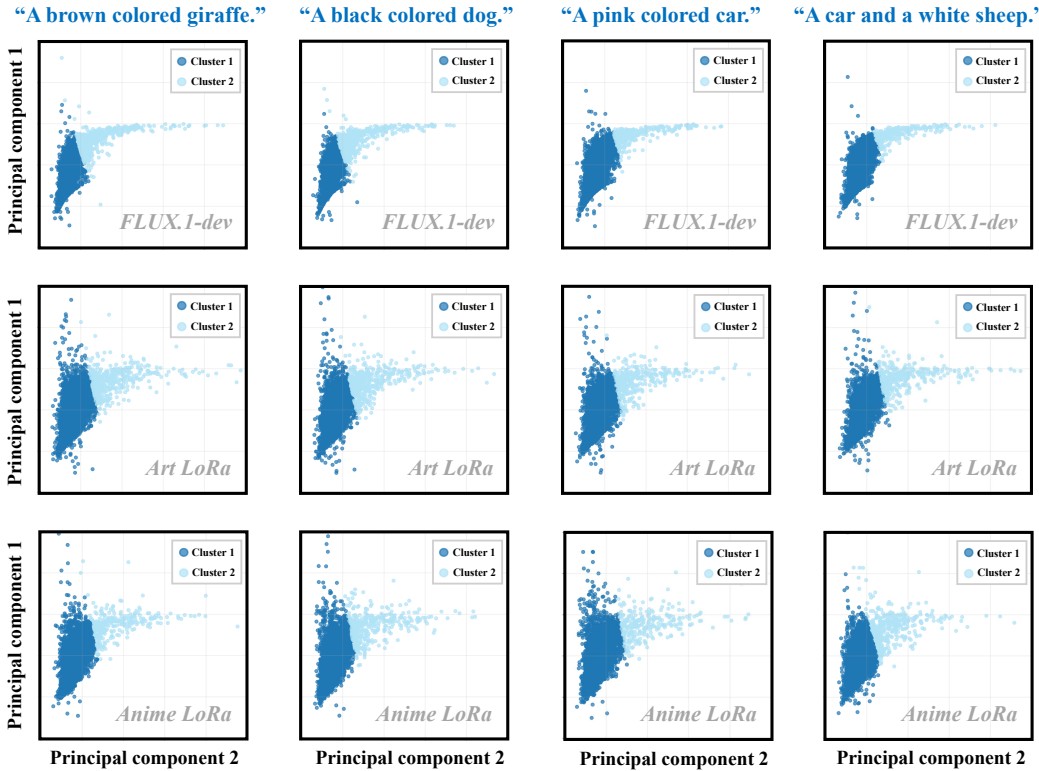

Figure 12: **Clustering consistency between original and LoRA-tuned models.** Each subplot visualizes the clustering results under different prompts for FLUX.1-dev (top row), and its two LoRA variants, the Art LoRA (middle row) and Anime LoRA (bottom row). Despite fine-tuning, the cluster boundaries remain highly consistent across prompts and models, indicating that HyCa's solver assignments remain stable and reusable even after LoRA adaptation.

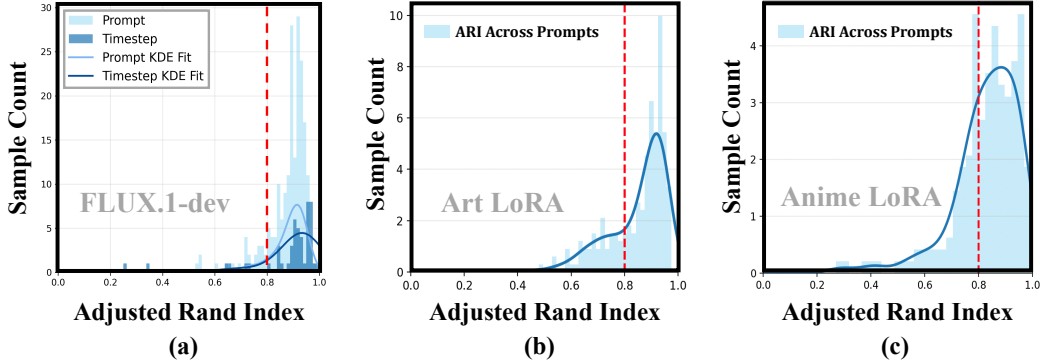

Figure 13: **ARI between clustering assignments from FLUX.1-dev and its LoRA variants.** (a) ARI distributions across different prompts and timesteps for FLUX.1-dev show clustering consistency within the base model. (b) and (c) compare the original model's clustering with that from Art LoRA and Anime LoRA, respectively. In both cases, the ARI values remain high (most above 0.8, marked by the red dashed line), confirming strong agreement between the LoRA and original clustering results. This further supports that solver assignments in HyCa can be reused across LoRA variants.

be re-computed. Since LoRA only updates a small subset of parameters through low-rank adapters, most feature dynamics and their clustering structures should remain stable.

We extract feature trajectories from both LoRA-tuned models and compare their clustering assignments with those from the original FLUX.1-dev model. As shown in Fig. 12, the clustering results remain visually consistent. We further quantify this similarity by computing the Adjusted Rand Index (ARI) between the LoRA cluster assignments and those of the original model. As shown in Fig. 13(b–c), in both cases, the mean ARI exceeds 0.8, indicating strong agreement and confirming that HyCa's clustering results are largely invariant under LoRA adaptation.

Therefore, we believe that for models sharing the same backbone, switching between different LoRA does not require re-clustering or solver re-selection in most practical cases. Admittedly, it is infeasible to exhaustively test all possible LoRA adapters; however, even in rare cases where HyCa may require re-clustering due to drastic adapter-induced changes, the offline process only needs about 1 second to assign new solvers for the updated LoRA. This demonstrates HyCa's robustness and efficiency in real-world deployment scenarios where multiple LoRA adapters are used for personalization or domain adaptation.

### A.6 DEMONSTRATION OF RUNGE-KUTTA METHOD

Our use of Runge–Kutta (RK) follows the standard ODE formulation and is then adapted to the discrete setting of feature caching.

**Classical Runge–Kutta method.** Consider a single feature dimension following an ODE

$$\frac{dy}{d\tau} = f(\tau, y), \qquad y(\tau_n) = y_n,$$

with step size $h = \tau_{n+1} - \tau_n$. A standard second order Runge–Kutta method computes:

$$k_1 = f(\tau_n, y_n), \tag{27}$$
$$k_2 = f(\tau_n + h, \, y_n + hk_1), \tag{28}$$

and updates:

$$y_{n+1} = y_n + \frac{h}{2}\big(k_1 + k_2\big). \tag{29}$$

Intuitively, Runge–Kutta averages the slope at the beginning and (predicted) end of the step to obtain a second order accurate approximation of the trajectory.

**Adapting RK2 to discrete feature caching.** In our setting we do not have direct access to $f(\tau, y)$, because evaluating it would require additional forward passes of the diffusion transformer. Instead, for each timestep index $t_n$ and feature dimension $d$, we only have the cached activations $F_{t_n}^{(d)}, F_{t_{n-1}}^{(d)}, F_{t_{n-2}}^{(d)}, \dots$ from previously computed steps. To construct an RK predictor that uses only cached features, we replace the derivatives in equation 29 with finite differences.

With uniform step size $h$, we approximate the instantaneous slopes by backward differences:

$$\Delta_n = \frac{F_{t_n}^{(d)} - F_{t_{n-1}}^{(d)}}{h}, \qquad \Delta_{n-1} = \frac{F_{t_{n-1}}^{(d)} - F_{t_{n-2}}^{(d)}}{h}.$$

Here $\Delta_n$ plays the role of $f(\tau_n, y_n)$, while an extrapolated slope

$$\widetilde{\Delta}_{n+1} = \Delta_n + \big(\Delta_n - \Delta_{n-1}\big)$$

approximates $f(\tau_n + h, y_n + hk_1)$ in equation 29. Substituting $k_1 \approx \Delta_n$ and $k_2 \approx \widetilde{\Delta}_{n+1}$ into equation 29 yields the following discrete RK2 predictor in feature space:

$$\widehat{F}_{t_{n+1}}^{(d)} = F_{t_n}^{(d)} + \frac{h}{2}\big(\Delta_n + \widetilde{\Delta}_{n+1}\big) \tag{30}$$

$$= F_{t_n}^{(d)} + \frac{h}{2}(\Delta_n + \Delta_n + (\Delta_n - \Delta_{n-1})) \tag{31}$$

$$= F_{t_n}^{(d)} + \frac{3h}{2}\Delta_n - \frac{h}{2}\Delta_{n-1} \tag{32}$$

$$= \tfrac{3}{2}F_{t_n}^{(d)} - \tfrac{1}{2}F_{t_{n-2}}^{(d)}. \tag{33}$$

This is exactly the form we use in HyCa for an RK2 solver: it operates purely on cached features and does not invoke any extra network forward passes. During the offline "one-time choosing" stage, such discrete RK2 predictors (along with AB, BDF, Taylor, etc.) are instantiated per cluster, their next-step prediction errors are measured on recorded trajectories, and the solver with the lowest error is assigned to that cluster. At inference time, the chosen predictor equation 33 is then applied to forecast $\widehat{F}_{t_{n+1}}$ from $\{F_{t_n}, F_{t_{n-2}}\}$, providing a concrete and fully specified example of how RK2 is used for feature prediction in our framework.

In practice, we instantiate $F_{t_n}^{(d)}$ as the residual output of a specific DiT block at timestep $t_n$, rather than as the full diffusion latent. At a compute step, we evaluate the original block function and obtain the exact residual update, which is then cached as $F_{t_n}$. At a skip step, we bypass the expensive block computation and instead use a discrete solver (e.g., the RK2 rule in equation 33) to predict the next residual from cached outputs, dimension-wise:

$$\widehat{F}_{t_{n+1}} \;=\; \mathrm{RK2}\big(F_{t_n}, F_{t_{n-1}}, F_{t_{n-2}}\big),$$

or analogously for AB, BDF, Taylor, and other solvers in our pool. The predicted residual $\widehat{F}_{t_{n+1}}$ is then injected back into the network via the standard residual connection of that block, i.e., the block output is obtained by adding $\widehat{F}_{t_{n+1}}$ to the block input hidden state, exactly as in the original DiT architecture. Chaining this operation over all blocks yields the final hidden representation at timestep $t_{n+1}$. Therefore, HyCa only substitutes the per-block residual computations at skip steps with solver-based predictions on cached features, while the residual pipeline and the outer diffusion sampler remain unchanged.

### A.7 CLUSTERING STABILITY UNDER EXTREME SETTINGS

Beyond the standard configurations reported in the main paper, we further evaluate the robustness of our clustering procedure under several extreme settings. Our goal is to verify that the "one-time choosing" assumption continues to hold even when the generation conditions deviate significantly from common practice.

**Extreme resolutions.** We first examine clustering stability under highly atypical resolutions. In particular, we test very high resolutions (e.g., $2048 \times 2048$) as well as extremely narrow aspect ratios (e.g., $128 \times 1024$). For each setting, we extract feature trajectories and compute clustering assignments using the same indicators and hyperparameters as in the default setting. Across all tested resolutions, the resulting cluster structures remain highly consistent: the same dimensions are assigned to the same clusters with only negligible variation, and the ARI between extreme and standard resolutions stays well above 0.8 which we consider stable, as shown in Fig.14(a).

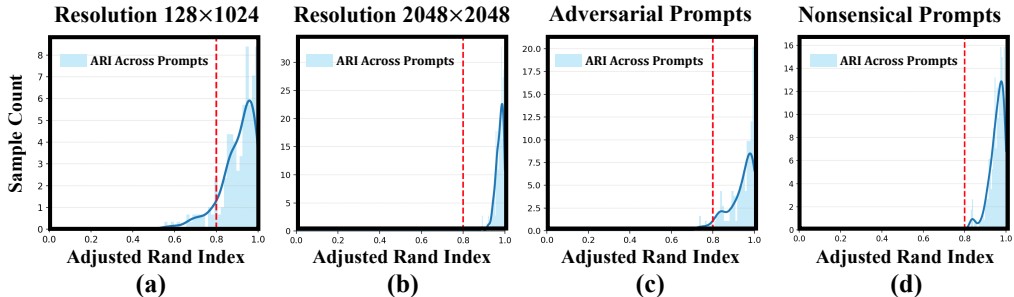

Figure 14: **(a–b)** The left two figures show that clustering structures remain stable under extreme resolution settings. **(c–d)** The right two figures show that clustering is also consistent for nonsensical prompts and closely matches those from normal, well-formed prompts.

**Adversarial and nonsensical prompts.** We also probe stability with respect to input text by using a diverse collection of prompts, including grammatically incorrect descriptions, deliberately ambiguous instructions, and even semantically nonsensical phrases. Despite the irregular semantics, the induced feature trajectories still yield cluster assignments that closely match those from normal, well-formed prompts. Example clustering results are shown in Fig.14(b).

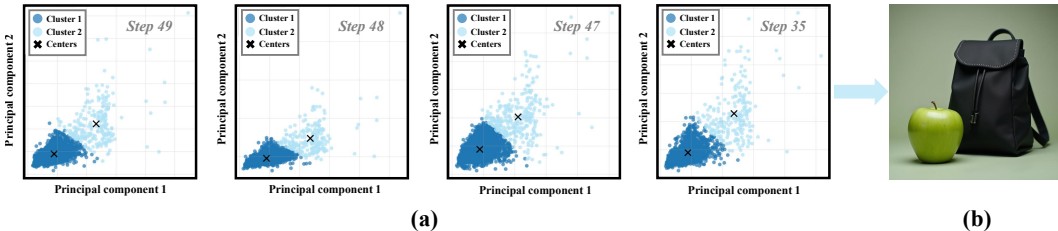

(a)                                                                                                (b)

Figure 15: **(a)** Although the clustering results at Step 49, Step 48, and Step 47 are inconsistent, the assignment at Step 47 already aligns closely with that at Step 35, suggesting that instability is limited to the first few timesteps in rare cases. **(b)** Despite this early-stage instability, the final generated image remains unaffected, as the first few steps are fully computed by the model and not involved in feature caching. Most predictors require at least three historical features, and caching typically begins at Step 4 or later—after cluster assignments have stabilized.

Other than the extreme cases discussed above, we further investigate how different timesteps influence the clustering behavior. Our analysis shows that clustering can exhibit mild instability only at very early denoising steps. The noticeable deviation occurs rarely in a small subset of cases at the very beginning of the denoising process. Specifically, we observe that the cluster assignments during the first $1-3$ timesteps can be less stable in some cases as shown in Fig.15(a). However, this does not affect the final generation quality. In actual inference, the first 3 steps are always computed fully by the model without caching, since most of our predictors require at least three historical features to make accurate forecasts. As a result, even if cluster assignments are slightly unstable during the initial steps, they have no impact on the quality or correctness of cached predictions. Feature caching actually begins after the first few compute steps, by which time the clustering results have already stabilized, therefore the generation quality is ensured as shown in Fig.15(b).

