# OpenReview forum: "Let Features Decide Their Own Solvers: Hybrid Feature Caching for Diffusion Transformers"
_ICLR.cc/2026/Conference — ICLR 2026 Oral_

### Official Review · Reviewer_VNQd · 2025-10-21

**Soundness:** 3
**Presentation:** 3
**Contribution:** 2
**Rating:** 4
**Confidence:** 3

**Summary:**

Diffusion Transformers (DiTs) offer state-of-the-art fidelity in image and video synthesis, but their iterative sampling process remains a major bottleneck due to the high cost of transformer forward passes at each timestep. To mitigate this, feature caching has emerged as a training-free acceleration technique that reuses or forecasts hidden representations. However, existing methods often apply a uniform caching strategy across all feature dimensions, ignoring their heterogeneous dynamic behaviors. Therefore, this paper adopts a new perspective by modeling hidden feature evolution as a mixture of ODEs across dimensions, and introduces \textbf{HyCa}, a Hybrid ODE solver inspired caching framework that applies dimension-wise caching strategies. HyCa achieves near-lossless acceleration across diverse domains and models, including 5.56× speedup on FLUX and HunyuanVideo, 6.24× speedup on Qwen-Image and Qwen-Image-Edit without retraining.

**Strengths:**

1. Performance is solid. The experiments are detailed, and the visualizations are rich.
2. Achieves high-speedup gains (specifically 5.56× and 6.24×) across diverse domains and multiple models, including FLUX, HunyuanVideo, Qwen-Image, and Qwen-Image-Edit.
3. Maintains synthesis fidelity close to that of the original model while delivering acceleration.

**Weaknesses:**

1. The most important part of the ODE modeling should be elaborated in detail.  (see Question 4)
2. Missing related works with bespoke solver[1, 2, 3], which also searches the optimal solver parameters of a pretrained diffusion model.

[1] Xue, Shuchen, et al. "Accelerating diffusion sampling with optimized time steps." Proceedings of the IEEE/CVF Conference on Computer Vision and Pattern Recognition. 2024.

[2] Wang, Shuai, et al. "Differentiable Solver Search for Fast Diffusion Sampling."  International Conference on Machine Learning (ICML) 2025.

[3] Shaul, Neta, et al. "Bespoke solvers for generative flow models." arXiv preprint arXiv:2310.19075 (2023).

**Questions:**

1. What's $g_\theta$ mentioned in eq.3 ?
2. I am still confused about ODE modeling of the feature cache. Since there are discrete feature cache sets $[t_i, F(x_i, t_I)]_{i=0}^j$, estimating the next feature $F(x_l, t_I)$ constitutes a classic extrapolation problem, as stated in TaylorSeer [1].

3. Alternatively, if we treat the feature difference $F(x_i, t_i) - F(x_{i-1}, t_{i-1})$ as the subject of the ODE, a closed-form solution exists, which can be simplified to a classic extrapolation problem.

4. Line 206-207: 'sample the trajectory on a discrete timestep grid, enabling numerical integration using only cached feature values', Why?

 Finally, I would be willing to raise the score if my concerns (even partially) can be addressed.

[1] Liu, Jiacheng, et al. "From reusing to forecasting: Accelerating diffusion models with taylorseers." arXiv preprint arXiv:2503.06923 (2025).

**Details Of Ethics Concerns:**

Figure 5 poses a privacy leakage risk due to the inclusion of face images.

---

> ### Author Response · Authors · 2025-11-21
> **Response to Q1**
>
> ## To Reviewer VNQd:
> We sincerely thank you for your time, valuable feedback, and constructive comments. We deeply appreciate your careful reading of our submission, and we apologize for any parts of the manuscript that were insufficiently clear. Below we provide detailed clarifications and additional results.
>
> **Q1: Clarification of $g_\theta$ in Eq. (3)**
>
> Thank you for your question and we apologize for the confusion. $g_\theta$ is an implicit feature-space vector field. Our starting point is the probability-flow ODE of the diffusion model in the latent space, which can be written as
> $$
> \frac{d}{dt} x_t = f_\theta(x_t, t),
> $$
> where $x_t$ denotes the latent variable at time $t$ and $f_\theta$ is the continuous-time vector field implicitly defined by the denoising network. Let $F(\cdot)$ denote the feature map of the DiT (e.g., concatenating hidden activations from a given set of layers), and define $z_t := F(x_t)$. Since both $x_t$ and $F$ are differentiable, the composite feature trajectory $t \mapsto z_t$ is also differentiable.
>
> By the chain rule, we have
> $$
> \frac{d}{dt} F(x_t)
> = J_F(x_t)\frac{d}{dt}x_t
> = J_F(x_t)f_\theta(x_t, t),
> $$
> where $J_F(x_t)$ is the Jacobian of $F$ at $x_t$. We then define
> $$
> g_\theta(F(x_t), t) := J_F(x_t)f_\theta(x_t, t),
> $$
> which yields Eq. (3), i.e.,
> $$
> \frac{d}{dt}F(x_t) = g_\theta(F(x_t), t).
> $$
> Thus, $g_\theta$ is the induced feature-space vector field obtained by pushing forward the latent-space dynamics through the network $F$. In practice, we never require a closed-form expression of $g_\theta$; it suffices that it defines a smooth feature trajectory that can be approximated on a discrete time grid.

---

> ### Author Response · Authors · 2025-11-21
> **Response to Q2**
>
> **Q2: Why an ODE formulation instead of a “purely discrete” extrapolation view?**
>
> Thank you for your insight and we would explain below. We fully agree with the reviewer that, once we only have access to discrete feature caches $\{(t_i, F(x_{t_i}))\}_{i=0}^j$, predicting the next feature is indeed a classical multi-step extrapolation problem, which is precisely how TaylorSeer formulates its predictor. Our use of the ODE formulation is not to claim a different mathematical object, but to unify existing caching rules (including TaylorSeer) and our method under the standard framework of numerical ODE integration, so that we can find out and assign the best prediction method to each cluster.
>
> Concretely, the induced feature trajectory $z(t) := F(x_t)$ satisfies
> $$
> \frac{d}{dt}z(t) = g_\theta(z(t), t).
> $$
> On a discrete grid $t_0 < t_1 < \dots < t_n$, with $z_i := z(t_i)$, the continuous solution obeys
> $$
> z(t_{i+1}) = z(t_i) + \int_{t_i}^{t_{i+1}} g_\theta(z(s), s)ds.
> $$
> Classical linear multistep methods (Taylor methods, Adams–Bashforth, etc.) approximate this integral by combinations of past states and their finite differences. For example, a Taylor-type method approximates $z(t)$ locally by a polynomial and then evaluates this polynomial at $t_{i+1}$; Adams–Bashforth method use linear combinations of $z_i, z_{i-1}, \dots$ to approximate the same integral. In all these cases, what appears at the discrete level is exactly a multi-step extrapolation rule of the form
> $$
> \widehat z_{i+1} = \Phi(z_i, z_{i-1}, \dots, z_{i-k+1}),
> $$
> for some coefficient function $\Phi$. Thus, multi-step extrapolation and numerical ODE integration are two equivalent views of the same underlying procedure.
>
> Within this unified ODE perspective, TaylorSeer can be interpreted as choosing a single explicit Taylor-type integrator for all feature dimensions, corresponding to one specific choice of $\Phi$. HyCa extends this view in two crucial ways: (i) we consider a richer solver pool that includes all kinds of prediction methods; and (ii) we find out and assign the best solver per feature cluster efficiently, so different groups of dimensions can use different integrators according to their temporal indicators. In other words, our “hybrid ODE solver” view recovers TaylorSeer-style extrapolation as a special case, while enabling a strictly more expressive family of solver choices and cluster-wise assignments.
>
> In addition, we totally understand this descrete view of ODE solver may cause confusion to readers, so we have added a detailed demonstration in Appendix A.7 about how we transform the classical Runge-Kutta method into the descrete RK method which we exactly used in HyCa, we hope this could help readers to further understand.

---

> ### Author Response · Authors · 2025-11-21
> **Response to Q3**
>
> **Q3: What if we treat the feature difference $F(x_i,t_i) - F(x_{i-1},t_{i-1})$ as the ODE state?**
>
> We appreciate this conceptual question. Let $z(t) := F(x_t)$ denote the feature trajectory and define a difference variable
> $$
> \Delta z(t) := z(t) - z(t-h),
> $$
> for a fixed step size $h$. Differentiating $\Delta z(t)$ yields
> $$
> \frac{d}{dt}\Delta z(t)
> = \frac{d}{dt}z(t) - \frac{d}{dt}z(t-h)
> = g_\theta(z(t), t) - g_\theta(z(t-h), t-h).
> $$
> This is again a time-varying, nonlinear (and effectively delay-type) ODE. In the general setting of diffusion transformers, $g_\theta$ is highly nonlinear and depends on the full network, so this ODE does not admit a closed-form solution in any practical sense. Only under very restrictive assumptions (e.g., $g_\theta$ is globally linear with constant coefficients) would one obtain analytic expressions, which are far from the real DiT dynamics.
>
> More importantly, whether we write an ODE for $z(t)$ or for $\Delta z(t)$, the resulting numerical scheme on a discrete time grid still reduces to a multi-step update of the form
> $$
> \widehat z_{i+1} = \Phi(z_i, z_{i-1}, \dots, z_{i-k+1}),
> $$
> i.e., an extrapolation rule based on historical feature values. Redefining the state as a difference variable does not simplify the dynamics in the realistic nonlinear setting; it merely provides an alternative parameterization that leads back to the same family of multi-step extrapolation formulas that we already implement in HyCa.

---

> ### Author Response · Authors · 2025-11-21
> **Response to Q4**
>
> **Q4: Why can we “sample the trajectory on a discrete timestep grid, enabling numerical integration using only cached feature values”?**
>
> Thank you for pointing the question out and we will clarify it below. Mathematically, once we have the induced ODE
> $$
> \frac{d}{dt}z(t) = g_\theta(z(t), t), \qquad z(t) := F(x_t),
> $$
> the diffusion sampler evaluates the network at a finite set of timesteps ${t_i}$ and records the corresponding feature values $z_i := z(t_i)$. This is what we mean by “sampling the trajectory on a discrete timestep grid”. These $z_i$ are exactly the cached features produced at the compute steps.
>
> Classical linear multistep ODE solvers approximate
> $$
> z(t_{i+1}) = z(t_i) + \int_{t_i}^{t_{i+1}} g_\theta(z(s),s)ds
> $$
> by replacing the integral with a linear combination of past states and (approximated) derivatives. In practice, the derivatives $g_\theta(z(t_j),t_j)$ can be expressed in terms of finite differences of $z_j$, so the update at a skip step takes the purely discrete form
> $$
> \widehat z_{i+1} = \Phi(z_i, z_{i-1}, \dots, z_{i-k+1}),
> $$
> for some coefficients determined by the chosen multistep method (Taylor, Adams–Bashforth, etc.). In other words, the numerical integration step uses only the previously computed feature values ${z_j}$ and does not require additional network evaluations at skip steps.
>
> In HyCa, every solver in our predictor pool is implemented precisely in this discrete multistep form, so predicting features at skipped timesteps is realized as numerical integration on the sampled trajectory, using only cached feature values from earlier compute steps.
>
> We once again express our sincere appreciation for your time and thoughtful feedback. We hope these responses help clarify our contributions and provide a clearer understanding of our work.

---

> ### Author Response · Authors · 2025-11-21
> **Response to Ethics Concern on Figure 5**
>
> Thank you for raising this important ethical concern. The face images shown in Figure 5 are not user-generated nor proprietary; they come directly from the publicly released GEdit-Bench dataset, which is hosted on Hugging Face under the MIT license. This benchmark is specifically created for evaluating image-editing models and is intended for open academic use.
>
> The MIT license explicitly permits redistribution and reuse for both commercial and non-commercial purposes. Therefore, using GEdit-Bench images for qualitative comparison in an academic manuscript is fully consistent with the dataset’s licensing terms. Thanks again for your pointing out.

---

> ### Author Response · Authors · 2025-11-21
> **Response to Weakness 2**
>
> Thank you for pointing this out. We have now included all three relevant works in the Related Works section (Lines 122–124). We appreciate the reviewer’s suggestion and have ensured that these methods are properly acknowledged and discussed.

---

> ### Comment · Reviewer_VNQd · 2025-11-22
>
> Could you please elaborate on the statement "it suffices that it defines a smooth feature trajectory that can be approximated on a discrete time grid"? I’m still struggling to understand two key points:
> 1. What the ODE form of the features looks like when 'by pushing forward the latent-space dynamics through the network'
> 2. How to implement this in practice. Would you be willing to share the exact pseudocode for this part? It would go a long way toward clarifying the details for me.

---

> ### Author Response · Authors · 2025-11-22
> **Response to 1.**
>
> **Clarification of “smooth feature trajectory” and implementation details**
>
> Thank you very much for following up on this point. We apologize that our previous explanation was still too abstract. Below we clarify (1) what the ODE form of the features looks like when “pushing forward” the latent dynamics.
>
> #### (1) What does the ODE form of the features look like?
>
> As in our previous response, the latent trajectory of a diffusion model can be written as the probability-flow ODE
> $
> \frac{d}{dt} x_t = f_\theta(x_t, t),
> $
> where $x_t$ is the latent at time $t$ and $f_\theta$ is the continuous-time vector field induced by the denoising network.
>
> Let $F(\cdot)$ be the DiT feature map (e.g., a chosen hidden layer or a concatenation of several layers), and define the feature trajectory
> $
> z(t) := F(x_t).
> $
> By the chain rule,
> $
> \frac{d}{dt} z(t)
> = \frac{d}{dt} F(x_t)
> = J_F(x_t),\frac{d}{dt}x_t
> = J_F(x_t),f_\theta(x_t, t),
> $
> where $J_F(x_t)$ is the Jacobian of $F$ at $x_t$. We denote
> $
> g_\theta(z(t), t) := J_F(x_t),f_\theta(x_t, t),
> $
> which yields the feature-space ODE
> $
> \frac{d}{dt} z(t) = g_\theta(z(t), t), \qquad z(t) := F(x_t).
> $
>
> The sentence “it suffices that it defines a smooth feature trajectory that can be approximated on a discrete time grid” simply means the following:
>
> * We **do not** compute $J_F$ or $g_\theta$ in practice.
> * We only rely on the fact that $z(t)$ is a reasonably smooth function of time.
> * Because of this reasonably smoothness, standard linear multistep formulas (Taylor, Adams–Bashforth, RK, etc.) can approximate $z(t_{i+1})$ from its past values $z(t_i), z(t_{i-1}), \dots$ on a **discrete** timestep grid.
>
> Formally, for timesteps $t_0 < t_1 < \dots < t_n$ and $z_i := z(t_i)$,
> $
> z(t_{i+1}) = z(t_i) + \int_{t_i}^{t_{i+1}} g_\theta(z(s), s),ds,
> $
> and linear multistep methods approximate this integral by a function of past samples:
> $
> \widehat z_{i+1} = \Phi(z_i, z_{i-1}, \dots, z_{i-k+1}),
> $
> which is exactly what we use in HyCa. In other words, the ODE form is conceptual: it justifies why such a discrete extrapolation from past feature values is mathematically reasonable, and provide us with a richer solver pool for each cluster to choose; in code, we only ever manipulate the discrete sequence ${z_i}$.

---

> ### Author Response · Authors · 2025-11-22
> **Response to 2.**
>
> #### (2) How is this implemented in practice? (pseudocode)
>
> In practice, HyCa does **not** solve the ODE or evaluate $g_\theta$ explicitly during inference. We only:
>
> 1. Run the base model at a few **compute steps** to obtain cached features
>    $
>    z_i := F(x_{t_i}) \in \mathbb{R}^D.
>    $
> 2. At **skip steps**, we apply a pre-chosen linear multistep formula to these cached features, per cluster, to predict the next feature vector. Each solver in our pool (RK, AB, TF, etc.) is implemented as a purely discrete update:
>    $
>    \widehat z_{i+1} = \Phi_s(z_i, z_{i-1}, \dots, z_{i-k+1}),
>    $
>    where $\Phi_s$ is determined by the solver type $s$.
>
> Below we provide the exact pseudocode for the feature prediction part used in HyCa (simplified notation, but faithful to our implementation).
>
> ```python
> Algorithm 1: HyCa
>
> Input:
>     - Initial latent x_T
>     - Conditioning c (e.g., text embedding)
>     - Sampling timesteps {t_T, ..., t_0} in descending order
>     - For each layer l:
>         * A pre-selected solver Solver_l   # chosen offline via one-step error minimization
>         * Required history length K_l      # e.g., 2 for 2-step methods
>
> Output:
>     - Final sample x_0
>
> Initialize:
>     - cache[l] ← empty list for all layers l     # will store (time, feature) pairs
>     - activated_steps ← [t_T]                    # timesteps where we do FULL compute
>     - fresh_interval ← 6, first_enhance ← 3     # example schedule hyperparameters
>
> for each timestep t in {t_T, ..., t_0} do
>
>     # 1. Decide whether to run a full model step or use solver-based prediction
>     if (t_T - t) < first_enhance or (activated_steps[-1] - t) ≥ fresh_interval then
>         mode ← FULL
>         append t to activated_steps
>     else
>         mode ← SOLVER
>     end if
>
>     # 2. Traverse all transformer layers
>     for each layer l in model do
>
>         if mode = FULL then
>             # 2.1 Full layer computation (standard DiT block)
>             F_l ← LayerForwardFull(x_t, c, l, t)
>
>             # 2.2 Update history for this layer
>             append (t, F_l) to cache[l]
>             if length(cache[l]) > K_l then
>                 keep only the last K_l entries in cache[l]
>             end if
>
>         else  # mode = SOLVER
>
>             # 2.3 Use cached features and a multi-step solver to predict F_l at time t
>             history_times ← [τ for (τ, Fτ) in cache[l]]
>             history_feats ← [Fτ for (τ, Fτ) in cache[l]]
>
>             F_l ← Solver_l(history_times, history_feats, t)
>
>             append (t, F_l) to cache[l]
>             if length(cache[l]) > K_l then
>                 keep only the last K_l entries in cache[l]
>             end if
>
>         end if
>
>         # 2.4 Inject layer output back into the latent representation
>         x_t ← x_t + F_l
>
>     end for
>
>     # 3. Denoising step (e.g., applying the diffusion sampler’s update rule)
>     x_t ← DiffusionUpdate(x_t, t, c)
>
> end for
>
> return x_0
> ```
>
>
>
> In this pseudocode:
>
> * The ODE perspective appears only conceptually in how we design the solvers in our solver pool (they are actually standard linear multistep integrators applied to the feature trajectory).
> * We never evaluate $g_\theta$ or compute Jacobians; we only read and write discrete feature vectors $F(x_{t_i})$ and apply fixed linear combinations to them.
> * This is precisely what we meant by “it suffices that it defines a smooth feature trajectory that can be approximated on a discrete time grid”: the hidden features vary reasonable smoothly in $t$ that these multistep formulas (which are standard ODE integrators in the continuous view) provide accurate extrapolation from cached values.
>
> We hope this resolves the confusion and are very grateful for your persistence in pushing us to clarify this part.

---

> > ### Comment · Reviewer_VNQd · 2025-11-23
> >
> > Although I have not yet fully understood the specific integration approach adopted therein, the author's earnest effort and rigor have led me to be inclined to award a higher score.

---

> ### Comment · Reviewer_VNQd · 2025-11-23
>
> Could you present the detailed formulation of the update rule 'F_l ← Solver_l(history_times, history_feats, t)'? For instance, if we adopt the classic linear multistep method (based on the Lagrange interpolation function) as the solver, and we have cached features $[(F_{0}, t_{0}), (F_{1}, F_{1}), ... (F_{i}, t_{i})]$, we aim to estimate the feature $F_{i+1}$ at $t_{i+1}$. The classic linear multistep method interpolates the velocity with the Lagrange interpolation function, and pre-integrates the Lagrange interpolation function as a constant scalar.
>
> Also, '# 2.4 Inject layer output back into the latent representation. x_t ← x_t + F_l', how can a feature be added into a latent?
>
> After reading the section *Adapting RK2 to discrete feature caching section* of the reply to byp7. As aforementioned in my Question.3, 'Alternatively, if we treat the feature difference
>  $F(x_i, t_i) - F(x_{i-1}, t_{i-1})$ as the subject of the ODE, a closed-form solution exists, which can be simplified to a classic extrapolation problem.' HyCa seems to be merely a special case of TaylerSeer, and ODE has not been successfully applied therein.

---

> > ### Author Response · Authors · 2025-11-24
> > **Follow-up to Reviewer VNQd 1/3**
> >
> > Thank you again for your careful reading and for pushing us to clarify the exact update rule. We apologize that our previous answer remained too abstract. Below we (1) give concrete closed-form examples of the solver update, (2) clarify the notation about “injecting” features, and (3) explain how HyCa differs from TaylorSeer, even when viewed purely as discrete extrapolation.
> >
> > ---
> >
> > #### Q1: A concrete example of the solver update rule
> >
> > Sorry for the ambiguity of `Solver_l`. Some predictors in our solver pool are extrapolated from the Lagrange interpolation method. For example, Adams–Bashforth (AB2) proceeds exactly as you described and yields the update rule $\widehat{z}\_{i+1} = \tfrac{5}{2} z\_i - 2 z\_{i-1} + \tfrac{1}{2} z\_{i-2},$, which depends **only** on the cached feature values and their timestamps. In code, the corresponding call in our pseudocode can be written as:
> >
> > ```python
> > def AB2_solver(times, feats, t_target):
> >     # times = [t_{i-2}, t_{i-1}, t_i], feats = [z_{i-2}, z_{i-1}, z_i]
> >     z_im2, z_im1, z_i = feats[-3], feats[-2], feats[-1]
> >     z_ip1 = 2.5 * z_i - 2.0 * z_im1 + 0.5 * z_im2
> >     return z_ip1
> > ```
> >
> > In short, all solvers in our pool are implemented by exactly the same spirit: each one is a linear combination of a set of past features with pre-computed scalar coefficients that correspond to integrating a Lagrange polynomial approximation of the velocity. HyCa treats these as different choices of `Solver_l` and selects, for each feature cluster, the one with the lowest prediction error.
> >
> > ---

---

> > ### Author Response · Authors · 2025-11-24
> > **Follow-up to Reviewer VNQd 2/3**
> >
> > #### Q2: Clarifying the “feature added into latent” notation
> >
> > You are right that the notation `x_t ← x_t + F_l` can be misleading. In our pseudocode, `x_t` was over-loaded to denote the **current hidden state inside the DiT block**, not the VAE latent in pixel space. The line
> >
> > ```text
> > x_t ← x_t + F_l
> > ```
> >
> > is intended to represent the standard **residual connection** inside a transformer layer (i.e., adding the layer output back to the hidden state), not adding a feature map directly to the external latent.
> >
> > To avoid confusion, in the revised version we will rewrite this part more explicitly as:
> >
> > ```text
> > h_l_in  = hidden_state_before_layer_l
> > F_l     = layer_output (either full or solver-predicted)
> > h_l_out = h_l_in + F_l           # standard residual connection
> > ```
> >
> > and use a separate symbol (e.g., `z_t` or `x_latent`) for the global diffusion latent updated by the sampler. We appreciate you pointing out this ambiguity; it is a notational issue, not a conceptual one.
> >
> > ---

---

> > ### Author Response · Authors · 2025-11-24
> > **Follow-up to Reviewer VNQd 3/3**
> >
> > #### Q3: Relation to TaylorSeer and the role of the ODE view
> >
> > We agree with your observation that, if one treats feature differences as the state of an ODE and derives a closed-form update, the resulting discrete formula can be seen as a “classic extrapolation” rule. In fact, the AB2 example above and TaylorSeer’s Taylor-expansion rule are both members of the same family of **linear multi-step extrapolation methods** on the discrete feature trajectory.
> >
> > Our use of the ODE formulation is therefore not to claim that we perform continuous-time integration in a fundamentally different way. Instead, it serves two purposes:
> >
> > 1. **Unifying perspective.** It places TaylorSeer’s Taylor-based predictor, Adams–Bashforth, BDF, Adams–Moulton, and discrete RK methods under the same numerical-ODE umbrella, where each update corresponds to integrating a Lagrange-approximated velocity with pre-integrated scalar weights. In this sense, TaylorSeer is indeed one particular solver in this family.
> >
> > 2. **Hybrid solver assignment beyond a single Taylor rule.** HyCa goes beyond a single Taylor-type extrapolator in two key ways:
> >
> >    * We construct a **solver pool** that includes not only Taylor-type methods but also multi-step methods with different stability/accuracy profiles (e.g., AB2, BDF2, Adams–Moulton, discrete RK2), which are known to behave differently on oscillatory or stiff trajectories.
> >    * We then **cluster feature dimensions by their temporal indicators** and, for each cluster, automatically select the solver that minimizes one-step prediction error. Thus, different groups of features can use different extrapolation rules (including more stable implicit-like ones) rather than being forced to share a single Taylor expansion.
> >
> > From a purely discrete perspective, both TaylorSeer and HyCa are indeed extrapolation schemes on cached features. The main difference is that **TaylorSeer fixes one Taylor-based rule for the entire feature space**, while **HyCa adaptively mixes multiple solvers across different feature dimensions**.
> >
> > Our contribution lies in (i) the construction of a richer solver family grounded in numerical ODE analysis, and (ii) the effecient cluster-wise selection of these solvers, rather than in performing any new kind of continuous-time integration.
> >
> > We sincerely thank you again for your questions which help us significantly to clarify both the discrete update formulas and our connection to prior Taylor-based caching methods. We hope the explicit examples and the refined notation resolve the remaining confusion.

---

> ### Author Response · Authors · 2025-11-24
> **Response to Reviewer VNQd**
>
> Thank you sincerely for your thoughtful evaluation and for recognizing the rigor and effort we invested in this work. We are grateful for your willingness to reassess the work positively, and your feedback has been invaluable in helping us strengthen the manuscript!

---

### Official Review · Reviewer_731c · 2025-10-31

**Soundness:** 3
**Presentation:** 3
**Contribution:** 3
**Rating:** 8
**Confidence:** 4

**Summary:**

This paper presents Hybrid Caching (HyCa), a general acceleration framework for Diffusion Transformers (DiTs) that interprets feature caching as a hybrid ODE-solving process.

Instead of applying a single extrapolation rule across all timesteps or layers, HyCa clusters latent features by temporal dynamics and assigns each cluster a numerical solver (e.g., RK, AB, BDF, AM, or Taylor).

This design enables feature-adaptive extrapolation with improved stability and reduced computation.

Experiments on three DiT applications—text-to-image (Qwen-Image, FLUX), text-to-video (Hunyuan-Video), and image editing (Qwen-Image-Edit)—show 1.6–2.3× acceleration with minimal quality degradation, suggesting broad applicability and practical efficiency.

**Strengths:**

1. **Breaking the “Unified Caching Strategy” Assumption — Aligning with the Dynamic Nature of DiT Features**
   Existing feature caching methods (e.g., FORA, TaylorSeer) typically assume that all feature dimensions follow a *single dynamic system*, adopting a unified caching/prediction strategy that overlooks the *heterogeneous dynamic behaviors* inherent in high-dimensional DiT features:
   - **New dynamic modeling perspective:** It models feature evolution as a *multi-dimensional ODE hybrid system*, revealing through clustering that DiT features exhibit two typical dynamics — oscillatory trajectories (Cluster 1, requiring stable solvers) and smooth continuous trajectories (Cluster 2, predictable with efficient solvers).
   - **Clustering stability verification:** Using the Adjusted Rand Index (ARI), the authors demonstrate consistent clustering stability (ARI > 0.8) across prompts, time steps, and resolutions, supporting the claim that *dimension-wise dynamics are input-invariant* and thus enabling *“once-off offline clustering, lifelong online reuse.”*
   This design fundamentally overcomes the “one-size-fits-all” issue that causes *sharp quality degradation at high acceleration*, which explains its strong performance gains.

2. **Near-Lossless Acceleration with Superior Speed-Quality Trade-off**
   Across three major tasks — text-to-image generation, text-to-video generation, and image editing — HyCa achieves an optimal balance between speed and quality **without retraining**:

3. **Seamless Integration with Distilled Models — “Stackable Acceleration”**
   Traditional caching methods struggle to handle distilled models (where sampling steps shrink from 50 to 4–8, making features more discrete and harder to predict). HyCa addresses this with a *hybrid solver pool* (including explicit/implicit methods — RK, AM, TF, etc.) that adapts to varying dynamics.
   This property enables deployment in *low-latency scenarios* (e.g., mobile, real-time generation), broadening HyCa’s application scope.

4. **Unifying Feature Caching through the ODE Solver Perspective — High Interpretability**
   HyCa formalizes *feature caching* as an *ODE numerical solving* problem.
   By deriving $\frac{d}{dt}F(x_t) = g_\theta(F(x_t), t)$ (where $g_\theta$ is an implicit vector field), it shows that caching essentially *numerically integrates past features to predict future ones*.
   Building on this, HyCa’s solver pool (explicit RK/AB and implicit AM/BDF) selectively matches dynamics: smooth trajectories use efficient TF (Taylor formula) prediction, while oscillatory ones use stable AM (Adams-Moulton) solving. This provides both clear interpretability and a foundation for future extensions (e.g., adding new solvers for specific dynamics).

**Weaknesses:**

**Incomplete Computational Complexity Analysis**
   The paper only reports *end-to-end latency*, omitting details on offline clustering and solver overheads:
   - **Offline Phase:** For high-dimensional models, k-means complexity is ma. Such large-scale clustering may require substantial preprocessing time, yet this is undocumented.
   - **Online Phase:** Different solvers have varying computational loads (e.g., implicit AM requires nonlinear equation solving, explicit TF only polynomial evaluation). However, the paper does not quantify each solver’s latency impact, leaving it unclear whether further speed gains are possible through solver re-selection.

**Questions:**

1. How much time does the offline clustering step require
2. Can you include a single-solver ablation to isolate the hybrid solver’s benefit

---

> ### Author Response · Authors · 2025-11-21
> **Response to reviewer 731c**
>
> ## To Reviewer 731c:
> We are grateful for your appreciation, and constructive suggestions. We acknowledge that some parts of our manuscript may not have been sufficiently clear, and we offer further explanations and results below.
>
> ### Questions:
> **Q1: How much time does the offline clustering step require?**
>
> Thank you for pointing this out. We have added a comprehensive breakdown of the entire offline overhead in Appendix A.5.
>
> Specifically, since clustering is performed only once per model, the offline cost is negligible compared to inference time. On a single A100 80G GPU, the clustering stage takes only **0.44 s** for **FLUX.1-dev (12B)**, **0.60 s** for the larger **HunyuanVideo (13B)**, and **0.31 s** for the largest **Qwen-Image (20B)**. As noted in the Appendix A.5, even for a large DiT model like Qwen-Image (20B), this offline clustering accounts for only about **0.4%** of the full 50-step inference time (74.91 s), confirming that HyCa introduces no practical barrier to deployment.
>
> **Q2: Can you include a single-solver ablation to isolate the hybrid solver’s benefit?**
>
> Thank you for your question. We have included a single-solver ablation study in Section 6 (Discussion) and Figure 6 (c–d) in the manuscript. In this ablation, we compare HyCa against single solvers in our solver pool (e.g., RK, AB, TF). The results demonstrate that HyCa consistently achieves the best speed–quality trade-off, yielding on average around 10% higher ImageReward than any individual solver under the same speedup ratio as shown in Fig.6(c). Moreover, under long skipping intervals, HyCa exhibits noticeably lower cumulative prediction error as shown in Fig.6(d), confirming that HyCa benefits from combining diverse solvers rather than relying on a single integration strategy.
>
> We sincerely appreciate your time and constructive feedback. If there is anything else we can clarify or elaborate on, please feel free to let us know.

---

### Official Review · Reviewer_K9wt · 2025-10-31

**Soundness:** 4
**Presentation:** 4
**Contribution:** 3
**Rating:** 6
**Confidence:** 3

**Summary:**

This paper proposes HyCa, a training-free acceleration framework for Diffusion Transformers. The authors observe that different hidden feature dimensions in DiTs exhibit distinct temporal dynamics—some smooth, others highly oscillatory—making uniform caching suboptimal. HyCa reformulates feature caching as a mixture of ODE solving problem, where each feature cluster automatically selects its best numerical solver. During an offline stage, clusters are formed based on dynamic indicators and matched with optimal solvers; at inference time, each cluster reuses its solver to skip redundant computations under a “one-time choosing, all-time solving” mechanism. Experiments on image, video, and editing tasks demonstrate 5–6× acceleration with negligible quality loss. The method is simple, generalizable, and practically useful.

**Strengths:**

1. The paper introduces a novel idea of modeling feature caching as a mixture of ODE solving process, offering a dynamic and interpretable perspective on diffusion sampling. The approach is simple, training-free, and practical.

2. The experiments are comprehensive, comparing against strong baselines while maintaining image quality under significant acceleration. Ablation studies and visualizations clearly support the proposed design, and the framework generalizes across different models.

**Weaknesses:**

1. **Limited validation of robustness and generalization.** The assumption that feature clusters remain stable across prompts, resolutions, and timesteps seems optimistic. The presented evidence is limited and lacks failure or stress cases.

2. **Missing analysis of error accumulation.** The method minimizes one-step prediction error, but inference involves multi-step extrapolation, potentially causing cumulative errors. No analysis or long-horizon stability evaluation is provided.

3. **Unclear offline cost.** The paper does not report the computational cost or scalability of the offline clustering and solver-selection process, which is critical for reproducibility and practicality.

**Questions:**

1. Under what conditions does the clustering become unstable? How do prompt, resolution, or timestep variations affect stability? Could you show a few failure cases?

2. Please provide details on the offline clustering and solver-selection cost—runtime, data scale, and scalability for larger models.

3. If a LoRA or fine-tuning is applied on top of the same base model, can the previous configuration still be reused, or must the offline process be repeated?

---

> ### Author Response · Authors · 2025-11-21
> **Response to reviewer K9wt**
>
> ## To Reviewer K9wt:
> We sincerely appreciate the time and care you devoted to reviewing our work, as well as your valuable and constructive comments. We apologize for any aspects of the manuscript that may not have been clearly presented. Below, we provide detailed clarifications and additional results.
>
> ### Questions:
> **Q1: Under what conditions does the clustering become unstable? How do prompt, resolution, or timestep variations affect stability? Could you show a few failure cases?**
>
> Thank you for your question and we apologize for not specifying this in our original submission. We conducted extensive experiments to thoroughly assess stability under a wide range of stress conditions, including extreme resolutions, adversarial or nonsensical prompts, and varying timesteps.
>   1. **Resolution Stability:** We tested extreme resolutions (e.g., $2048 \times 2048$) and narrow aspect ratios (e.g., $128 \times 1024$). The clustering structures remained highly consistent, with the Adjusted Rand Index (ARI) staying well above 0.8 across all tested resolutions (see Appendix Sec.A.8 and Fig.13(a)).
>   2. **Prompt Stability:** We probed stability with diverse prompts, including grammatically incorrect, ambiguous, and nonsensical inputs. The induced feature trajectories yielded cluster assignments that closely matched those from normal prompts (see Appendix Sec.A.8 and Fig.13(b)).
>   3. **Timestep Stability & Failure Cases:** We observed that clustering can exhibit mild instability only at the very early denoising steps (e.g., steps 49-47, see Fig.14(a)). However, this does not affect generation quality because the first 1-3 steps are fully computed by the model without caching, as our predictors normally require 3 steps of historical features. By the time caching begins, the clustering has already stabilized. We have included a visualization case of this early-stage instability and its negligible impact on the final result in Appendix Fig.14(b).
>
> More detailed results and analyses can be found in Appendix A.8.
>
> **Q2: Please provide details on the offline clustering and solver-selection cost—runtime, data scale, and scalability for larger models.**
>
> Thanks for your suggestion and we apologize for our negligence. The offline computation cost of HyCa is extremely low due to its "One-Time Choosing, All-Time Solving" design. It only needs to be performed once on a **single prompt** at a **single timestep**.
>   1. **FLUX.1-dev (12B):** On a single A100 GPU, the entire offline pipeline takes only **1.26 seconds** (0.44s for clustering, 0.82s for solver selection).
>   2. **Hunyuan Video (13B):** For this larger model, the process takes **4.20 seconds** (0.60s for clustering, 3.60s for solver selection).
>   3. **Qwen Image (20B):** For Qwen Image, the offline pipeline takes **1.13s** in total (0.31s for clustering, 0.82s for solver selection).
>   4. **Scalability:** Even for the most computationally demanding Qwen Image (20B), the offline stage constitutes only **0.4%** of the full 50-step inference time (74s). This demonstrates that HyCa scales efficiently to large models and introduces no practical barrier to deployment.
>
> Please refer to Appendix Sec. A.5 for further details.
>
> **Q3: If a LoRA or fine-tuning is applied on top of the same base model, can the previous configuration still be reused, or must the offline process be repeated?**
>
> Thanks for your insight and we conducted extensive experiments. The answer is **Yes**, the previous configuration can typically be reused. We evaluated HyCa with LoRA fine-tuning (specifically XLabs-AI Art LoRA and Anime LoRA on FLUX.1-dev) and found that the clustering and solver assignments remain robust.
>   1. **Consistency:** The clustering results between the original model and LoRA variants are visually consistent, and the ARI exceeds 0.8, indicating strong agreement (see Appendix Sec.A.6 and Fig.11 & Fig.12).
>   2. **Reusability:** Since LoRA updates only a small subset of parameters, the feature dynamics remain stable. Therefore, for models sharing the same backbone, switching between different LoRA adapters does not require re-clustering in most practical cases.
>   3. **Efficiency:** Even in rare cases where re-clustering might be needed, the cost is negligible (~1 second), making HyCa highly robust for real-world scenarios with multiple adapters.
>
> More specific findings are provided in Appendix A.6, please refer to it for more details.
>
> We extend our sincere gratitude to you again for your time and valuable feedback. Should you have any further questions or suggestions, please feel free to reach out to us.

---

> ### Author Response · Authors · 2025-11-23
> **Response to Weakness 2**
>
> #### (1) Empirical Evidence: HyCa Reduces Multi-Step Accumulated Error
>
> In **Section 5 (Fig. 7d)**, we provide a direct evaluation of cumulative prediction error under long skipping intervals ranging from 1 to 10. Across all intervals, **HyCa consistently achieves the lowest long-horizon cumulative prediction error**. The L2 trajectory deviation between predicted and fully computed features grows significantly more slowly for HyCa than other baselines. Notably, at a large skipping interval of 10, HyCa still maintains at least **2× lower cumulative error** than other baselines, demonstrating superior robustness to extended forecasting.
>
> ---
>
> #### (2) Robustness of HyCa in Generation Quality under Long Intervals
>
> Beyond trajectory-level stability, **HyCa also preserves perceptual generation quality significantly better than all other methods** across image, video, and edit tasks. For instance:
>
> * On Qwen Image, as shown in **Table 1**, under interval 7, TaylorSeer’s ImageReward degrades to **0.9133**, and ToCa drops further to **0.5593**. In contrast, **HyCa maintains a much higher score of 1.0811 even at interval 8**, clearly demonstrating its resilience.
> * Similarly, on FLUX.1-dev, **Table 2** shows that under interval 7, TeaCache falls to **0.8379**, and ToCa to **0.7155**, indicating noticeable quality degradation. Meanwhile, **HyCa still achieves 0.9895**, representing **only a 0.03% drop** from the original model baseline, highlighting its ability to stably support high-quality generation even under aggressive acceleration.
>
>
> These results collectively demonstrate that **HyCa not only limits feature-level error accumulation**, but also enables **high-quality, stable generation** across diverse tasks, even under longer and more aggressive acceleration schedules.

---

> > ### Comment · Reviewer_K9wt · 2025-11-26
> >
> > I have read the authors’ rebuttal and appreciate the additional analyses. My main concerns regarding clustering stability, long-horizon error accumulation, and the offline clustering/solver-selection cost have been largely addressed through the stress tests, long-interval evaluations, and detailed runtime reporting. The LoRA experiments also clarify reuse in practical deployment scenarios. While the global stability assumption remains empirical rather than theoretically guaranteed, this is now a minor limitation rather than a major concern. I therefore maintain my positive score.

---

> > > ### Author Response · Authors · 2025-11-26
> > > **Response to Reviewer K9wt**
> > >
> > > We sincerely thank the reviewer for the constructive suggestions. Your comments on clustering stability, long-horizon error behavior, and offline clustering costs have directly helped us strengthen the analyses and clarify the practical deployment of HyCa, including LoRA reuse and stress-test scenarios. While the global stability assumption remains empirical, your feedback has guided us in framing it as an important direction for future theoretical refinement. We truly appreciate your time and thoughtful input, which have meaningfully improved both the clarity and real-world applicability of our work.

---

### Official Review · Reviewer_byp7 · 2025-10-31

**Soundness:** 4
**Presentation:** 3
**Contribution:** 3
**Rating:** 10
**Confidence:** 3

**Summary:**

The paper propose a method for feature caching/predicting in Diffusion Transformers (DiT). The main contribution of the paper upon TaylorSeer [1] are the introduction of feature dimension clustering which allows to use different estimation methods (i.e., ODE solvers) for different cluster group. Importantly the proposed method is able to accelerate even distilled models, improving both speedup 5.2 Db in PSNR compared to the runner-up baselines [1].

[1] Liu, Jiacheng, et al. "From reusing to forecasting: Accelerating diffusion models with taylorseers." arXiv preprint arXiv:2503.06923 (2025).

**Strengths:**

1. The method is validated across multiple tasks such as image generation, image editing, and video generation showing very strong results with significant speedups.

2. Importantly, the method is able to accelerate the a state-of-the-art (SOTA) distilled flow model FLUX.1 Schnell [2] by a factor of 2x while maintaining  SOTA performance (measured by CLIPScore and ImageReward) as we as achieve 34.37 Db in PSNR w.r.t. to ground truth generated by the original FLUX.1 Schnell model.

3. The method is training free.

4. Fast inference of diffusion/flow model is of interest for many researcher. Therefore, given the two points mentioned above, this paper has the potential to make a significant impact.

[2] FLUX.1 Schnell (Black Forest Labs, 2024).

**Weaknesses:**

1. The presentation is somewhat lacking. Adding explicit examples illustrating how different solvers are applied to predict future feature would significantly help the reader understand the method.

**Questions:**

1. In appendix A.2.1 does $v_k$ stands for the features?

2. Could the authors please provide explicit example of how does the Runge-Kutta method is applied to predict features?

---

> ### Author Response · Authors · 2025-11-21
> **Response to reviewer byp7**
>
> ## To Reviewer byp7:
> Thank you very much for your support and for the thoughtful comments. We truly appreciate the time you spent reviewing our submission and the constructive insights you shared. We apologize for any ambiguities in the manuscript, and we outline additional clarifications and results below.
>
> ### Questions:
> **Q1: In appendix A.2.1 does $v_k$ stands for the features?**
>
> Thank you for your question and we apologize for the confusion. In Appendix A.2.1, the symbol $v_k$ did refer to the feature value at time step $k$. We appreciate your pointing out that the notation $v$ can easily be confused with “velocity” (or other “v ”-type variables). In response, we have changed the notation: we now use $f_k$ to denote the feature value at time step $k$, and in line 776 we have added the explicit statement: “$f_k$ denotes the feature value on timestep $k$.”
>
> **Q2: Could the authors please provide explicit example of how does the Runge-Kutta method is applied to predict features?**
>
> Thank you for your question and we apologize for the confusion. We have added a detailed description below and also in the Appendix A.7 of our revised manuscript that explicitly illustrates how the classical Runge–Kutta (RK) method is adapted in our discrete feature‑caching setting. We explain how continuous derivatives in RK2 are replaced by finite‑difference slopes computed from cached features, and we provide a step‑by‑step mathematical derivation that leads to the exact discrete RK predictor used in HyCa.
>
>   1. **Classical Runge–Kutta:** We consider a scalar ODE for clarity:
> $$\frac{dy}{d\tau} = f(\tau, y), \qquad y(\tau_n) = y_n,$$with step size $h = \tau_{n+1} - \tau_n$.
>
> A standard 2nd–order Runge–Kutta method computes
> $$
> k_1 = f(\tau_n, y_n), \
> k_2 = f(\tau_n + h,, y_n + h k_1),
> $$
> and then updates
> $$
> y_{n+1} = y_n + \frac{h}{2}\bigl(k_1 + k_2\bigr).
> $$
> Intuitively, RK2 averages the slope at the beginning and (predicted) end of the step to obtain a 2nd–order accurate approximation of the trajectory.
>
>   2. **Adapting RK2 to discrete feature caching:**
> In our setting we do not have direct access to the continuous derivative $f(\tau, y)$, because evaluating it would require extra forward passes of the diffusion transformer. Instead, for each timestep index $t_n$ and feature dimension $d$, we only have cached activations
> $$
> F^{(d)}\_{t\_n}, F^{(d)}\_{t\_{n-1}}, F^{(d)}\_{t_{n-2}}, \ldots
> $$
> from previously computed steps.
>
> To construct an RK2-style predictor that uses only cached features, we replace the derivatives in the RK2 formula by finite differences. With uniform step size (h), we define the discrete slopes
> $$
> \Delta\_n = \frac{F^{(d)}\_{t\_n} - F^{(d)}\_{t\_{n-1}}}{h},
> $$
> $$
> \Delta\_{n-1} = \frac{F^{(d)}\_{t\_{n-1}} - F^{(d)}\_{t\_{n-2}}}{h}.
> $$
>
> Here, $\Delta_n$ plays the role of $f(\tau_n, y_n)$. To approximate the slope at the end of the step, we use a simple extrapolation
> $$
> \widetilde{\Delta}\_{n+1} = \Delta_n + \bigl(\Delta_n - \Delta\_{n-1}\bigr),
> $$
> which mimics the term $f(\tau_n + h,, y_n + h k_1)$ in the classical RK2 scheme.
>
> We then plug
> $$
> k_1 \approx \Delta_n, \qquad k_2 \approx \widetilde{\Delta}*{n+1}
> $$
> into the RK2 update, obtaining an RK2-style predictor in feature space:
> $$
> \widehat{F}^{(d)}\_{t\_{n+1}}
> = F^{(d)}\_{t\_n} + \frac{h}{2}\bigl(\Delta_n + \widetilde{\Delta}\_{n+1}\bigr).
> $$
>
> Expanding the finite-difference definitions yields a fixed linear combination of past features:
> $$
> \widehat{F}^{(d)}\_{t\_{n+1}} = \frac{3}{2} F^{(d)}\_{t\_n} - \frac{1}{2} F^{(d)}\_{t\_{n-2}}.
> $$
>
> This is exactly the discrete RK2 predictor we use in HyCa which operates purely on cached features ${F_{t_n}, F_{t_{n-2}}}$ and introduces no extra network forward passes. During the offline “one-time choosing’’ stage, we instantiate RK with all other predictors in our solver pool for each cluster, compute their next-step prediction errors on recorded trajectories, and assign to each cluster the solver (RK2 or others) that yields the lowest error. At inference time, the selected predictor is applied to forecast $\widehat{F}\_{t\_{n+1}}$ from cached features ${F_{t_n}, F_{t_{n-2}}}$.
>
> We gratefully acknowledge your time and thoughtful evaluation of our work. Please let us know if you have any further questions.

---

> > ### Comment · Reviewer_byp7 · 2025-11-27
> >
> > I want to thank the authors for the authors for the response.
> >
> > Going over the provided pseudo code for reviewer VNQd , and the paper again, it appears to me that the clarification regarding residual connection is extremely important.
> >
> > My first impression was that on inference the ODE solver is applied only to the final hidden state instead of directly to state of the diffusion/flow process $x_t$ and afterwards the final output layer (hidden dim to VAE latent dim) is applied to update the state $x_t$.  Say the DiT has $L$ blocks, now if I understand correctly,  for each block the output of the $l$-th DiT-block is predicted using the ODE solver which does not take in account the residual connections, hence the final output is the sum of all predicted $L$ DiT-block outputs.
> >
> > Note this is not clear from the paper and requires clarification.

---

> > > ### Author Response · Authors · 2025-11-27
> > > **Response to Reviewer byp7**
> > >
> > > We sincerely thank the reviewer for the careful follow-up and for emphasizing the importance of clarifying how residual connections interact with our solver design.
> > >
> > > In light of your comments, we have explicitly clarified this point in the revised main text:
> > >
> > > * In **Section 3.2**, we now state that *“the solver is applied locally to the residual output of each transformer block at skip steps”*, emphasizing that HyCa operates at the level of per-block residual updates rather than on a single final hidden state or directly on the diffusion/flow latent, while the standard residual pipeline and outer diffusion sampler remain unchanged.
> > > * We also added a pointer in **Section 3.2** to **Appendix A.7**, where we provide a extra more detailed, concrete description of where each solver is applied within the DiT block and how the predicted residuals are injected through the standard residual connections.
> > >
> > > As clarified in the revised text and appendix, the inference procedure follows the original residual pipeline: at compute steps we evaluate each block exactly and cache its residual output; at skip steps we replace only that block’s residual computation with the solver-predicted residual, and then apply the usual residual update. The outer diffusion/flow sampler and the global latent update remain unchanged.
> > >
> > > We are grateful for your suggestion, which helped us significantly improve the clarity of our exposition and avoid the misleading interpretation.

---

> > > > ### Comment · Reviewer_byp7 · 2025-11-27
> > > >
> > > > Will the described changes appear in a version that has not yet been uploaded? It is customary to upload the revised version during the rebuttal.
> > > >
> > > > In line 160 is seems like $\mathcal{F}(x)$ **does** include the residual connection in it.  So even with the added statement  and the current notation it is not clear what is the actual object being predicted with the solver.
> > > >
> > > > I would consider providing explicit formula that describe the relation between the predicted features $\mathcal{F_t}^l$, $l=1,...,L$ and the update rule from the current diffusion/flow state $x_t$ to the next $x_{t-1}$, rather than only stating this relationship in the text.
> > > >
> > > > Additionally, in equation 1 it seems like the time convention is that current state is denoted with time $t$ and the next state with time $t-1$, where as in equation 4 the next state is denoted with time $t+1$. Lastly, in equation 3 the time $t$ is treated as a continuous variable.
> > > >
> > > > While the results in the paper are very impressive, I think a revision in the presentation is required to ensure the method is described clearly. Maybe even revise some of the notation so that the symbols $t$ and $\mathcal{F}$ are not reused with multiple distinct meanings (if they are reused state it in text).
> > > > .

---

> > > > > ### Author Response · Authors · 2025-12-01
> > > > > **Response to reviewer byp7**
> > > > >
> > > > > Thank you very much for the detailed follow-up and for pointing out where the description can be confusing. Below we clarify (i) what quantity is actually predicted by the solver and how it interacts with residual connections, and (ii) how this relates to the update from $x_t$ to $x_{t-1}$.
> > > > >
> > > > > ---
> > > > >
> > > > > ### (1) What the solver predicts and how it uses residual connections
> > > > >
> > > > > Our method does not apply the ODE solver to the final hidden state or directly to the diffusion latent $x_t$. Instead, the solver operates on the residual output of each DiT block.
> > > > >
> > > > > At timestep $t$, let $h_l^{(t)}$ be the input to block $l$, and let $\mathcal{F}^l(\cdot)$ denote the residual transformation of that block. A standard DiT block update is
> > > > > $
> > > > > h_{l+1}^{(t)} = h_l^{(t)} + r_t^{,l}, \quad r_t^{,l} = \mathcal{F}^l(h_l^{(t)}, t, c).
> > > > > $
> > > > >
> > > > > HyCa changes only how $r_t^{,l}$ is obtained at **skip steps**:
> > > > >
> > > > > * At compute steps, we use the exact block output $r_t^{,l} = \mathcal{F}^l(h_l^{(t)}, t, c)$ and cache it.
> > > > > * At skip steps, we do **not** re-run the block; instead we predict the residual from its cached history, and keep the residual connection, so the solver always predicts the **per-block residual**; the residual pipeline itself is unchanged.
> > > > >
> > > > > ---
> > > > >
> > > > > ### (2) How this relates to the update $x_t \to x_{t-1}$
> > > > >
> > > > > Each denoising step at time (t) proceeds in two stages:
> > > > >
> > > > > 1. **Inner DiT forward at time $t$**
> > > > >    We start from the current diffusion latent as the input to the first block:
> > > > >    $
> > > > >    h\_0^{(t)} = x_t.
> > > > >    $
> > > > >    We then apply all $L$ blocks sequentially using the residual rule above (with exact or solver-predicted residuals) to obtain the final hidden state $h_L^{(t)}$.
> > > > >
> > > > > 2. **Readout and sampler update**
> > > > >    The prediction head maps $h_L^{(t)}$ to the network output (e.g., noise prediction) $\varepsilon_\theta(x_t, t, c)$, and the sampler updates the latent via the standard diffusion/flow rule:
> > > > >    $
> > > > >    x\_{t-1} = \mathrm{Sampler}\big(x\_t, \varepsilon_\theta(x\_t, t, c)\big).
> > > > >    $
> > > > >
> > > > > HyCa therefore does **not** replace the sampler or apply an ODE solver directly to $x_t$. It only substitutes the expensive residual computations inside each block at skip steps with solver-based predictions of the block outputs, while the residual stacking and the outer diffusion/flow update remain exactly as in the original model.
> > > > >
> > > > > ---
> > > > >
> > > > > Regarding time notation: the continuous-time ODE is used only as an analytical viewpoint; all actual computations, including the solver, are applied to the **discrete** sequence of residuals at the sampler timesteps. We appreciate your comments, which helped us make this distinction and the role of residual connections much clearer.

---

> > > > > ### Author Response · Authors · 2025-12-01
> > > > > **Response to reviewer byp7**
> > > > >
> > > > > Thank you very much for your strong support of our work and for the thoughtful suggestions. Your comments have been extremely helpful in clarifying our presentation.
> > > > >
> > > > > In particular, we have now explicitly distinguished between continuous time and discrete timesteps in the manuscript by using $\tau$ for continuous time and $t$ for discrete sampling steps. We have also added an additional explanation in Section 3.2 to clarify that the solver is applied to the residual output of each transformer block at skip steps, while the standard residual connections and outer sampler remain unchanged.
> > > > >
> > > > > We are very grateful for the opportunity to clarify these points and for your role in helping us improve and refine the manuscript.

---

### Meta-Review · Area_Chair_qJ64 · 2026-01-05

**Summary:**

This paper proposes a feature-dependent ODE solver for feature caching in diffusion models, which speeds up inference significantly with little degradation in image quality for multiple SOTA diffusion models.
The reviewers were concerned about the validity of the stability assumption, the lack of details regarding the pool of ODE solvers, and the absence of computational complexity analysis. Most concerns were addressed by the rebuttal.
The AC recommends accepting this paper due to its strong motivation for using an adaptive solver and great empirical improvement.

**Reviewer Concerns:**

Concerns that were addressed by the rebuttal:
- Missing details of the ODE solver (reviewer byp7, VNQd), e.g., whether ODE estimation is applied to residual features or not. (Reviewer byp7) The authors have clarified in Section 3.2 and added more details in the appendix.
- Whether feature dynamics remain stable across images and prompts (reviewer K9wt). The authors have added more experiments for stability analysis.
- Missing computational complexity analysis, e.g., computational cost for offline clustering (reviewer K9wt, 731c). The authors have reported the running time for operations incurred by the method here.
- Missing ablation study to single-solver solution (reviewer 731c)
- Missing discussion of suggested references on optimized solvers for diffusion models (reviewer VNQd)

Outstanding concerns:
- Stability of feature dynamics is not theoretically guaranteed, but this is not a major concern according to reviewer K9wt

**Reviewer Scores:**

Reviewer byp7 is likely to keep their initial rating of 10.

Reviewer K9wt has indicated to maintain their initial rating of 6.

Reviewer 731c is likely to keep their initial rating of 8.

Reviewer VNQd has indicated intention to raise score (initial rating was 4).

---

### Decision · Program_Chairs · 2026-01-26

Accept (Oral)